

# Evidence from IASI of a speeding up in stratospheric O₃ recovery in the Southern Hemisphere contrasting with a decline in the Northern Hemisphere

Catherine Wespes[1], Daniel Hurtmans[1], Simon Chabrillat[2], Gaétane Ronsmans[1], Cathy Clerbaux[3,1] and Pierre-François Coheur[1]

[1] Université Libre de Bruxelles (ULB), Faculté des Sciences, Chimie Quantique et Photophysique, Bruxelles, Belgique
[2] Belgian Institute for Space Aeronomy, Brussels, Belgium
[3] LATMOS/IPSL, Sorbonne Université, UVSQ, CNRS, Paris, France

**Abstract**

In this paper, we present the global fingerprint of recent changes in the mid-upper stratospheric (MUSt; >25hPa) ozone ($O_3$) in comparison with the lower stratospheric (LSt, 150-25 hPa) $O_3$ derived from the first 10 years of the IASI/Metop-A satellite measurements (January 2008 – December 2017). The IASI instrument provides vertically-resolved $O_3$ profiles with very high spatial and temporal (twice daily) samplings, allowing to monitor $O_3$ changes in these two regions of the stratosphere. By applying multivariate regression models with adapted geophysical proxies on daily mean $O_3$ time series, we discriminate anthropogenic trends from various modes of natural variability, such as the El Niño/Southern Oscillation – ENSO. The representativeness of the $O_3$ response to its natural drivers is first examined. One important finding relies on a pronounced contrast between a positive LSt $O_3$ response to ENSO in the extra-tropics and a negative one in the tropics, with a delay of 3 months, which supports a stratospheric pathway for the ENSO influence on lower stratospheric and tropospheric $O_3$. In terms of trends, we find an unequivocal $O_3$ recovery from the available period of measurements in winter/spring at mid-high latitudes for the two stratospheric layers sounded by IASI (>~35°N/S in the MUSt and >~45°S in the LSt) as well as in the total columns at southern latitudes (>~45°S) where the increase reaches its maximum. These results confirm the effectiveness of the Montreal protocol and its amendments, and represent the first detection of a significant recovery of $O_3$ concurrently in the lower, in the mid-upper stratosphere and in the total column from one single satellite dataset. A significant decline in $O_3$ at northern mid-latitudes in the LSt is also detected, especially in winter/spring of the northern hemisphere. Given counteracting trends in LSt and MUSt at these latitudes, the decline is not categorical in total $O_3$. When freezing the regression coefficients determined for each natural driver over the whole IASI period but adjusting a trend, we calculate a significant speeding up in the $O_3$ response to the decline of $O_3$ depleting substances (ODS) in the total column, in the LSt and, in a lesser extent, in the MUSt, at high southern latitudes over the year. A significant acceleration of the $O_3$ decline at northern mid-latitudes in the LSt and in the total column is also highlighted over the last years. That, specifically, needs urgent investigation for identifying its exact origin and apprehending its impact on climate change.



## 1 Introduction

Ozone is a key radiatively active gas of the Earth atmosphere, in both the troposphere and the
stratosphere. While in the troposphere, $O_3$ acts as a strong pollutant and an important greenhouse
gas, in the stratosphere, and more particularly in the middle-low stratosphere, it forms a
protective layer for life on Earth against harmful solar radiation. In the 1980s, the scientific
community motivated decision-makers for regulating the use of CFCs, after the unexpected
discovery of the Antarctic ozone hole (Chubachi, 1984; Farman et al., 1985) induced by
continued use of chlorofluorocarbons (CFC's; Molina and Rowland, 1974; Crutzen,1974); These
latter are at the origin of the massive destruction of $O_3$ following heterogeneous reactions on the
surface of polar stratospheric clouds (Solomon, 1999). The world's nations reacted to that
human-caused worldwide problem by ratifying the International Vienna Convention for the
Protection of the Ozone Layer in 1985 and the Montreal Protocol in 1987 with its later
amendments, which forced the progressive banning of these ozone depleting substances (ODS)
in industrial applications by early 1990s with a total phase-out of the most harmful CFCs by the
year 2000.

Ozone is very sensitive to changes in (photo-)chemistry, therefore a recovery from $O_3$ depletion
is expected in response to the Montreal Protocol and its amendments, but with a delayed period
due to the long residence time of halocarbons in the atmosphere (Hofmann et al., 1997; Dhomse
el al., 2006; WMO, 2007; 2011). The decline in CFCs was only initiated about 10 years after
their phasing out (Anderson et al., 2000; Newman et al., 2006; Solomon et al., 2006 ; Mäder et
al., 2010; WMO, 2011; 2014). The early signs of ozone response to that decline were confirmed
in several studies that reported first a slowdown in stratospheric ozone depletion (e.g.
Newchurch et al., 2003; Yang et al., 2008), followed by a leveling off of upper stratospheric (e.g.
WMO, 2007) and total $O_3$ (e.g. WMO, 2011; Shepherd et al., 2014) depletion since the 2000's.
A significant onset of recovery was identified later for upper stratospheric $O_3$ (e.g. WMO, 2014;
2018; Harris et al., 2015). Only a few studies have shown evidence for increasing total column
$O_3$ in polar region during springtime (e.g. Salby et al., 2011; Kuttippurath et al., 2013; Shepherd
et al., 2014; Solomon et al., 2016). No reliable estimates of long-term trend in total $O_3$ columns
(TOC) at global scale have been reported yet, likely because of counteracting trends in the
different vertical atmospheric layers. Ball et al. (2018) have found that a continuing $O_3$ decline
prevails in the low stratosphere since 1998, leading to a slower increase in total $O_3$ than expected
from the effective equivalent stratospheric chlorine (EESC) decrease. However, the reported
decline is not reproduced by the state-of-the-art models and its exact reasons are still unknown
(Ball et al., 2018). Galytska et al. (2019) recently suggested that the decline is dynamically
controlled by variations in the tropical upwelling.



Although recent papers based on observational datasets and statistical approaches agree that we
currently progress towards an emergence into ozone recovery (e.g. Pawson et al., 2014; Harris et
al., 2015; Steinbrecht et al., 2017; Sofieva et al., 2017; Ball et al., 2018; Weber et al., 2018),
trend magnitude and trend significance over the whole stratosphere substantially differ from one
study to another and, consequently, they are still subject to controversy (Keeble et al., 2018). A
clear identification of the onset of $O_3$ recovery is very sensitive due to concurrent sources of $O_3$
fluctuations (e.g. Reinsel et al., 2005; WMO, 2007, 2011). They include: changes in solar
ultraviolet irradiance, in atmospheric circulation patterns such as the quasi-biennial oscillation
(QBO; Baldwin et al., 2001) and the El Niño–Southern Oscillation (ENSO; e.g. Randel et al.,
2009), in temperature, in ODS emissions and volcanic eruptions (e.g. Mt Pinatubo in 1991 and
Calbuco in 2015) with their feedbacks on stratospheric temperature and dynamics (e.g. Jonsson
et al., 2004). Furthermore, the differences in vertical/spatial resolution and in retrieval
methodologies (inducing biases), possible instrumental degradations (inducing drifts), and use of
merged datasets into composites, likely explain part of the trend divergence between various
studies. Merging may be performed on deseasonalized anomalies, which offers the advantage of
removing instrumental biases between the individual data records (Sofieva et al., 2017) but large
differences remain in anomaly values between the independent datasets, as well as large
instrumental drifts and drift uncertainty estimates that prevent deriving statistically accurate
trends (Harris et al., 2015; Hubert et al., 2016).
In this context, there is a pressing need for long-duration, high-density and homogenized $O_3$
profile dataset to assess significant $O_3$ changes in different parts of the stratosphere and their
contributions to the total $O_3$.
In this paper, we exploit the high frequency (daily) and spatial coverage of the IASI satellite
dataset over the first decade of the mission (January 2008 – December 2017) to determine global
patterns of reliable trends in the stratospheric $O_3$ records, separately in the Mid-Upper
Stratosphere (MUSt) and the Lower Stratosphere (LSt). This study is built on previous analysis
of stratospheric $O_3$ trends from IASI, estimated on latitudinal averages over a shorter period
(2008-2013) (Wespes et al., 2016). A multivariate linear regression (MLR) model (annual and
seasonal formulations) that is similar to that previously used for tropospheric $O_3$ studies from
IASI (Wespes et al., 2017; 2018), but adapted here for the stratosphere with appropriate drivers,
is applied on gridded daily mean $O_3$ time series in the MUSt and the LSt. The MLR model is
evaluated in terms of its performance and of its ability to capture the observed variability in
Section 2, in terms of representativeness of $O_3$ drivers in Section 3 and in terms of adjusted
trends in Section 4. The minimum numbers of years of IASI measurements that is required to
indeed detect the adjusted trends from MLR in the two layers is also estimated in Section 4 that
ends with an evaluation of the trends detectable in polar winter and spring and with an evaluation
of a speeding up in the $O_3$ changes.





## 2 Dataset and methodology

### 2.1 IASI O$_3$ data

The Infrared Atmospheric Sounding Interferometer (IASI) is a nadir-viewing Fourier transform spectrometer designed to measure the thermal infrared emission of the Earth-atmosphere system between 645 and 2760 cm$^{-1}$. Measurements are taken from the polar sun-synchronous orbiting meteorological Metop series of satellites, every 50 km along the track of the satellite at nadir and over a swath of 2200 km across track. With more than 14 orbits a day and a field of view of four simultaneous footprints of 12 km at nadir, IASI provides global coverage of the Earth twice a day at about 09:30 AM and PM mean local solar time.

The Metop program consists of a series of three identical satellites successively launched to ensure homogenous measurements covering more than 15 years. Metop-A and –B have been successively launched in October 2006 and September 2012. The third and last satellite was launched in November 2018 onboard Metop-C. In addition to its exceptional spatio-temporal coverage, IASI also provides good spectral resolution and low radiometric noise, which allows the measurement of a series of gas-phase species and aerosols globally (e.g. Clerbaux et al., 2009; Hilton et al., 2012; Clarisse et al., 2018).

For this study, we use the O$_3$ profiles retrieved by the Fast Optimal Retrievals on Layers for IASI (FORLI-O$_3$; version 20151001) near-real time processing chain set up at ULB (See Hurtmans et al, 2012 for a description of the retrieval parameters and the FORLI performances). The FORLI algorithm relies on a fast radiative transfer and a retrieval methodology based on the Optimal Estimation Method (Rodgers, 2000) that requires a priori information (a priori profile and associated variance-covariance matrix). The FORLI-O$_3$ a priori consists of one single profile and one covariance matrix built from the global Logan/Labow/McPeters climatology (McPeters et al., 2007). The profiles are retrieved on a uniform 1 km vertical grid on 41 layers from surface to 40 km with an extra layer from 40 km to the top of the atmosphere considered at 60 km. Previous characterization of the FORLI-O$_3$ profiles (Wespes et al., 2016) have demonstrated a good vertical sensitivity of IASI to the O$_3$ measurement with up to 4 independent levels of information on the vertical profile in the troposphere and the stratosphere (MUSt; LSt; upper troposphere-lower stratosphere – UTLS – 300-150 hPa; middle-low troposphere – MLT – below 300 hPa). The two stratospheric layers that show distinctive patterns of O$_3$ distributions over the IASI decade (Fig.1a) are characterized by high sensitivity (DOFS > 0.85; Fig.1b) and low total retrieval errors (<5%; see Hurtmans et al., 2012; Wespes et al., 2016). The decorrelation between the MUSt and the LSt is further evidenced in Fig.1d that shows low correlation coefficients (< 0.4) between the mean absolute deseasonalized anomalies (as calculated in Wespes et al., 2017) in the two layers (Fig. 1c). Note that the highest correlation coefficients over the Antarctic (~0.4) are due to the smaller vertical sensitivity of the IASI measurements over cold surface (Clerbaux



et al., 2009). The latest validation exercises for the FORLI-O₃ product have demonstrated a high
degree of precision with excellent consistency between the measurements taken from the two
IASI instruments on Metop-A and -B, as well as a good degree of accuracy with biases lower
than 20% in the stratospheric layers (Boynard et al., 2018; Keppens et al. 2017). Thanks to these
good IASI-FORLI performances, large-scale dynamical modes of O₃ variations and long-term O₃
changes can be differentiated in the four retrieved layers (Wespes et al., 2016). The recent
validations have, however, reported a drift in the MUSt FORLI-O₃ time series from comparison
with O₃ sondes in the northern hemisphere (N.H.) (~3.53±3.09 DU.decade⁻¹ on average over
2008–2016; Boynard et al., 2018) that was suggested to result from a pronounced discontinuity
("jump") rather than from a progressive change. Further comparisons with CTM simulations
from the Belgian Assimilation System for Chemical ObsErvations (BASCOE; Huijnen et al.,
2016; Chabrillat et al., 2018) confirm this jump that occurred on 15 September 2010 over all
latitudes (see Fig. S1 of the supplementary materials). The discontinuity is suspected to result
from updates in level-2 temperature data from Eumetsat that are used as inputs into FORLI (see
Hurtmans et al., 2012). The apparent drift reported by Boynard et al. (2018) results from the
jump and contrasts with a progressive "instrumental" drift. This is verified by the absence of drift
in the O₃ time series after the jump (non-significant drift of -0.38±2.24 DU.decade⁻¹ on average
over October 2010 – May 2017; adapted from Boynard et al., 2018). This is in line with the
excellent stability of the IASI Level-1 radiances over the full IASI period (Buffet et al., 2016).
From the IASI-BASCOE comparisons, the amplitude of the jump has been estimated as lower
than 2.0 DU in the 55°S–55°N latitude band and 4.0 DU in the 55°–90° latitude band of each
hemisphere. The effect of the jump on the calculation of significant trends derived in Section 4 is
found small enough to explain the trend, therefore, this estimated jump is not taken into account
in the MLR. The jump values will be, however, considered in the discussion of the O₃ trends
(Section 4).
Finally, the present study only uses the daytime measurements (defined with a solar zenith angle
to the sun < 83°) from the IASI-A (aboard Metop-A) instrument that fully covers the first decade
of the IASI mission. The daytime measurements are characterized by a higher vertical sensitivity
(e.g. Clerbaux et al., 2009). Quality flags developed in previous IASI studies (e.g. Boynard et al.,
2018) were applied a posteriori to exclude data with a poor spectral fit, with less reliability or
with cloud contamination.
**2.2 Multivariate regression model**
In order to unambiguously discriminate anthropogenic trends in O₃ levels from the various
modes of natural variability (illustrated globally in Fig.1c as deseasonalized anomalies), we have
applied to the daily MUSt and LSt O₃ time series, a MLR model that is similar to that previously
developed for tropospheric O₃ studies from IASI (see Eq. 1 and 2 in Wespes et al., 2017; 2018)
but here adapted to fit the stratospheric variations. In addition to harmonic terms that represent



the 1-yr and 6-month variations, the MLR model includes the anthropogenic $O_3$ response through
a linear trend (LT) term and a set of explanatory variables (commonly called "proxies") to
parameterize the geophysical processes influencing the abundance of $O_3$ in the stratosphere. The
MLR uses an iterative stepwise backward elimination approach to retain, at the end of the
iterations, the most relevant proxies (with a 95% confidence level) explaining the $O_3$ variations
(e.g. Mäder et al., 2007). Table 1 lists the selected proxies, their sources and their temporal
resolutions. The proxies describe the influence of the Quasi-Biennial Oscillation (QBO; visible
from the deseasonalized anomaly maps in Fig.1c with a typical band-like pattern around the
Equator) at 10 hPa and 30 hPa, of the North Atlantic and the Antarctic Oscillations (NAO and
AAO), of the El Niño/Southern Oscillation (ENSO), of the volcanic aerosols (AERO) injected
into the stratosphere, of the strength of the Brewer-Dobson circulation (BDC) with the Eliassen-
Palm flux (EPF), of the polar $O_3$ loss driven by the volume of polar stratospheric clouds (VPSC),
of the tropopause height variation with the geopotential height (GEO) and of mixing of
tropospheric and stratospheric air masses with the potential vorticity (PV). The main proxies in
terms of their influence on $O_3$ are illustrated in Fig. 2 over the period of the IASI mission. The
construction of the EPF, VPSC and AERO proxies, which are specifically used in this study, is
explained hereafter, while the description of the other proxies can be found in previous IASI
studies (Wespes et al., 2016; 2017).
The EPF proxy consists of the normalized upward component of the EP flux crossing 100 hPa
and spatially averaged over the 45°-75° latitude band for each hemisphere. The fluxes are
calculated from the NCEP/NCAR 2.5°x2.5° gridded daily reanalysis (Kalnay et al., 1996) over
the IASI decade. The VPSC proxy is based on the potential volume of PSCs given by the volume
of air below the formation temperature of nitric acid trihydrate (NAT) over 60°-90° north and
south and calculated from the ERA-Interim reanalysis and from the MLS climatology of nitric
acid (I. Wohltmann, private communication; Wohltmann et al., 2007; and references therein).
The PSC volume is multiplied by the EESC to account for the changes in the amount of
inorganic stratospheric chlorine that activates the polar ozone loss. The $O_3$ build-up and the polar
$O_3$ loss are highly correlated with wintertime accumulated EP flux and PSC volume, respectively
(Fusco and Salby, 1999; Randel et al., 2002; Fioletov and Shepherd, 2003 and Rex et al., 2004).
These cumulative EP flux and PSC effects on $O_3$ levels are taken into account by integrating the
EPF and VPSC proxies over time with a specific exponential decay time according to the
formalism of Brunner et al. (2006; see Eq. 4). We set the relaxation time scale to 3 months
everywhere, except during the wintertime build-up phase of $O_3$ in the extratropics (from October
to March in the N.H. and from April to September in the southern hemisphere - S.H.) when it is
set to 12 months. For EFP, it accounts for the slower relaxation time of extratropical $O_3$ in winter
due to its longer photochemical lifetime. For VPSC, the 12-month relaxation time accounts for a
stronger effect of stratospheric chorine on spring $O_3$ levels: the maximum of the accumulated
VPSC (Fig. 2) coincides with the maximum extent of $O_3$ hole that develops during springtime
and that lasts until November. Note that correlations between VPSC and EPF are possible since



the same method is used to build these cumulative proxies. VPSC and EPF are also dynamically
anti-correlated to some extent since a strong BDC is connected with warm polar stratospheric
temperatures and, hence, reduced PSC volume (e.g. Wohltmann et al., 2007).

The AERO proxy is derived from aerosol optical depth (AOD) of sulfuric acid only. That proxy
consists of latitudinally averaged (22.5°N-90°N – AERO-N, 22.5°S-90°S – AERO-S and 22.5°S-
22.5°N – AERO-Eq) extinction coefficients at 12 µm calculated from merged aerosol datasets
(SAGE, SAM, CALIPSO, OSIRIS, 2D-model-simulation and Photometer; Thomason et al.,
2018) and vertically integrated over the two IASI stratospheric $O_3$ columns (AERO-MUSt and
AERO-LSt). Fig.2 shows the AERO proxies (AERO-N, AERO-S and AERO-Eq) corresponding
to the AOD over the whole stratosphere (150-2 hPa), while Fig.3 represents the latitudinal
distribution of the volcanic sulfuric acid extinction coefficients integrated over the whole
stratosphere (top panel) and, separately, over the MUSt (middle panel) and the LSt (bottom
panel) from 2005 to 2017. The AOD distributions indicate the need for considering one specific
AERO proxy for each latitudinal band (AERO-N, AERO-S and AERO-Eq) and for each vertical
layer (AERO-MUSt and AERO-LSt). Note that, as an alternative proxy to AERO, the surface
area density of ambient aerosol, that represents the aerosol surface available for chemical
reactions, has been tested, giving similar results.

Note also that, similarly to what has already been found for tropospheric $O_3$ from IASI (Wespes
et al., 2016), several time-lags for ENSO (1-, 3- and 5-month lags; namely, ENSO-lag1, ENSO-
lag3 and ENSO-lag5) are also included in the MLR model to account for a possible delay in the
$O_3$ response to ENSO at high latitudes.

Finally, autocorrelation in the noise residual ε (t) (see Eq. 1 in Wespes et al., 2016) is accounted
for in the MLR analysis with time lag of one day to yield the correct estimated standard errors
for the regression coefficients. They are estimated from the covariance matrix of the regression
coefficients and corrected at the end of the iterative process by the autocorrelation of the noise
residual. The regression coefficients are considered significant if they fall in the 95% confidence
level (defined by 2σ level). In the seasonal MLR, the main proxies ($x_j X_{norm,j}$; with $x_j$, the
regression coefficient and $X_{norm,j}$ the normalized proxy) are replaced by four explanatory
variables ($x_{spr} X_{norm,spr} + x_{sum} X_{norm,sum} + x_{fall} X_{norm,fall} + x_{wint} X_{norm,wint}$) for each grid cell (see
Section 2.2 in Wespes et al., 2017). Hence, the seasonal MLR adjusts 4 coefficients (instead of
one in the annual MLR) to account for the seasonal $O_3$ response to changes in the proxy. If that
method avoids to over-constrain the adjustment by the year-round proxies and, hence, reduces
the systematic errors, the smaller daily data points covered by the seasonal proxies translate to a
lower significance of these proxies. This is particularly true for EPF and VPSC that compensate
each other by construction. As a consequence, the annual MLR is performed first in this study



and, then, complemented with the seasonal one when it is found helpful for further interpreting
the observations.

Figure 4 shows the latitudinal distributions of the $O_3$ columns in the two stratospheric layers over
the IASI decade (first panels in Fig.4 a and b), as well as those simulated by the annual MLR
regression model (second panels) along with the regression residuals (third panels). The root
mean square error (*RMSE*) of the regression residual and the contribution of the MLR model into
the IASI $O_3$ variations (calculated as $\dfrac{\sigma\left(O_3^{\text{Fitted\_model}}(t)\right)}{\sigma\left(O_3(t)\right)}$ where σ is the standard deviation relative
to the regression model and to the IASI time series; bottom panels) are also represented (bottom
panels). The results indicate that the model reproduces ~25-85% and ~35-95% of the daily $O_3$
variations captured by IASI in the MUSt and the LSt, respectively, and that the residual errors
are generally lower than 10% everywhere for the two layers, except for the spring $O_3$ hole region
in the LSt. The *RMSE* relative to the IASI $O_3$ time series are lower than 20 DU and 15 DU at
global scale in the LSt and the MUSt, respectively, except around the S.H. polar vortex in the
LSt (~30 DU). On a seasonal basis (figure not shown), the results are only slightly improved: the
model explains from ~35-90% and ~45-95% of the annual variations and the *RMSE* are lower
than ~12 DU and ~23 DU everywhere, in the MUSt and the LSt, respectively. These results
verify that the MLR models (annual and seasonal) reproduce well the time evolution of $O_3$ over
the IASI decade in the two stratospheric layers and, hence, that they can be used to identify and
quantify the main $O_3$ drivers in these two layers (see Section 3).

The MLR model has also been tested on nighttime FORLI-$O_3$ measurements only and
simultaneously with daytime measurements, but this resulted in a lower quality fit, especially in
the MUSt over the polar regions. This is due to the smaller vertical sensitivity of IASI during
nighttime measurements, especially over cold surface, which causes larger correlations between
stratospheric and tropospheric layers (e.g. 40-60% at high northern latitudes versus ~10-20% for
daytime measurements based on deseasonalized anomalies) and, hence, which mixes
counteracted processes from these two layers. For this reason, only the results for the MLR
performed on daytime measurements are presented and discussed in this paper.

**3 Drivers of $O_3$ natural variations**

Ascribing a recovery in stratospheric $O_3$ to a decline in stratospheric halogen species requires
first identifying and quantifying natural cycles that may produce trend-like segments in the $O_3$
time series, in order to prevent any misinterpretation of those segments as signs of $O_3$ recovery.
The MLR analysis performed in Section 2.2 that was found to give a good representation of the
MUSt and LSt $O_3$ records shows distinctive relevant patterns for the individual proxies retained
in the regression procedure, as represented in Fig. 5. The fitted drivers are characterized by



significant regional differences in their regression coefficients with regions of in-phase relation
(positive coefficients) or out-of-phase relation (negative coefficients) with respect to the IASI
stratospheric $O_3$ anomalies. The areas of significant drivers (in the 95% confidence limit) are
surrounded by non-significant cells when accounting for the autocorrelation in the noise residual.
Figures 6 a and b respectively represent the latitudinal distribution of the fitted regression
coefficients for the proxies showing latitudinal variation only in the $O_3$ response (namely, QBO,
EPF, VPSC, AERO and ENSO) and of the contribution of these drivers into the $O_3$ variability
(calculated as the product of the $2\sigma$ variability of each proxy by its corresponding fitted
coefficient, i.e. the $2\sigma$ variability of the adjusted signal of the proxies). The $2\sigma$ $O_3$ variability in
the IASI measurements and in the fitted MLR model are also represented (black and grey lines,
respectively). Figure 7 displays the same results as Fig. 6b but for the austral spring and winter
periods only (using the seasonal MLR).
The PV and GEO proxies are generally minor components (not shown here) with relative
contributions smaller than 10% and large standard errors (>80%), except in the tropics where the
contribution for GEO reaches 40% in the LSt due to the tropopause height variation. Each other
adjusted proxy (QBO, SF, EPF, VPSC, AERO, ENSO, NAO and AAO) is an important
contributor to the $O_3$ variations, depending on the layer, region, and season as described next:
1. QBO - The QBO at 10hPa and 30hPa are important contributors around the Equator for
the two stratospheric layers. It shows up a typical band-like pattern of high positive
coefficients confined equatorward of ~15°N/S where the QBO is known to be a dominant
dynamical modulation force associated with strong convective anomalies (e.g. Randel
and Wu, 1996; Tian et al., 2006; Witte et al., 2008). In that latitude band, QBO10 and
QBO30 explain up to ~8 DU and ~5 DU, respectively, of the MUSt and LSt yearly $O_3$
variations (see Fig. 5 and 6b; i.e. relative contributions up to ~50% and ~40% for
QBO10/30 in MUSt and LSt $O_3$, respectively). The QBO is also influencing $O_3$ variations
poleward of 60°N/S with a weaker correlation between $O_3$ and equatorial wind anomalies
as well as in the sub-tropics with an out-of-phase transition. That pole-to-pole QBO
influence results from the QBO-modulation of extra-tropical waves and its interaction
with the BDC (e.g. Fusco and Salby, 1999). A pronounced seasonal dependence is
observed in the out-of-phase sub-tropical $O_3$ anomalies in the MUSt, with the highest
amplitude oscillating between the hemispheres in their respective winter (~5 DU of $O_3$
variations explained by QBO10/30 at ~20°S during JJA and at ~20°N during DJF; see
Fig. 7b for the JJA period in the MUStn the DJF period is not shown), which is in
agreement with Randel and Wu (1996). The amplitude of the QBO signal is found to be
stronger for QBO30 than for QBO10 in the LSt, which is in good agreement with studies
from other instruments for the total $O_3$ (e.g. Baldwin et al., 2001; Steinbrecht et al., 2006;
Frossard et al., 2013; Coldewey-Egbers et al., 2014) and from IASI in the troposphere
(Wespes et al., 2017). The smaller amplitude of $O_3$ response to QBO10 in the LSt



compared to the MUSt is again in agreement with previous studies that reported changes
in phase of the QBO10 response as a function of altitude with a positive response in the
upper stratosphere and destructive interference in the mid-low stratosphere (Chipperfield
et al., 1994; Brunner et al., 2006).
2. SF - In the MUSt layer, the solar cycle $O_3$ response is one of the strongest contributors
and explains globally between ~2 and 15 DU of in-phase $O_3$ variations (i.e. higher $O_3$
records during maximum solar irradiance) with the largest amplitude over the highest
latitude regions (see Fig. 5; relative contribution up to ~20%). The solar influence in LSt
is more complex with regions of in-phase and out-of-phase $O_3$ variations. The impact of
solar variability on stratospheric $O_3$ abundance is due to a combination of processes: a
modification in the $O_3$ production rates in the upper stratosphere induced by changes in
spectral solar irradiance (e.g. Brasseur et al., 1993), the transport of solar proton event-
produced $NO_y$ from the mesosphere down to the mid-low stratosphere where it decreases
active chlorine and bromine and, hence, $O_3$ destruction (e.g. Jackman et al., 2000; Hood
and Soukharev, 2006; and references therein) and its impact on the lower stratospheric
dynamics including the QBO (e.g. Hood et al., 1997; Zerefos et al., 1997; Kodera and
Kuroda, 2002; Hood and Soukharev, 2003, Soukharev and Hood, 2006). As for the QBO,
the strong SF dependence at polar latitudes in the LSt with zonal asymmetry in the $O_3$
response reflects the influence of the polar vortex strength and of stratospheric warmings,
and are in good agreement with previous results (e.g. Hood et al., 1997; Zerefos et al.,
1997; Labitzke and van Loon, 1999; Steinbrecht et al., 2003; Coldewey-Egbers et al.,
2014).  It is also worth noting that because only one solar cycle is covered, the QBO and
SF effects could not be completely separated because they have a strong interaction
(McCormack et al., 2007).
3. EPF - The vertical component of the planetary wave Eliassen-Palm flux entering the
lower stratosphere corresponds to the divergence of the wave momentum that drives the
meridional residual Brewer-Dobson circulation. In agreement with previous studies (e.g.
Fusco and Salby, 1999; Randel et al., 2002; Brunner et al., 2006), fluctuations in the
BDC are shown to cause changes on stratospheric $O_3$ distribution observed from IASI:
EPF largely positively contributes to the LSt $O_3$ variations at high latitudes of both
hemispheres where $O_3$ is accumulated because of its long chemical lifetime, with
amplitude ranging between ~20 and 100 DU (see Fig. 5 and 6; i.e. relative contribution of
~35-150%). The influence of the EPF decreases at lower latitudes where a stronger
circulation induces more $O_3$ transported from the tropics to middle-high latitudes and,
hence, a decrease in $O_3$ levels particularly below 20 km (Brunner et al., 2006). The
influence of EP fluxes in the Arctic is the smallest in summer (see Fig.7; <~35 DU *vs* ~70
DU in fall; the two other seasons are not shown) due to the later $O_3$ build-up in polar
vortices. In the S.H., because of the deployment of the $O_3$ hole, the EP influence is



smaller than in the N.H. and the seasonal variations are less marked. In the MUSt, the $O_3$
response attributed to variations in EPF is positive in both hemispheres, with a much
lower amplitude than in the LSt (up to ~20-35 DU). The region of out-of-phase relation
with negative EPF coefficients over the high southern latitudes (Fig. 5b) is likely
attributable to the influence of VPSC that has correlations with EPF by construction (see
Section 2.2).
4. VPSC - Identically to EPF, VPSC is shown to mainly contribute to $O_3$ variations in LSt
over the polar regions (~55 DU or 40% in the N.H. *vs* ~60 DU or 85% in the S.H. on a
longitudinal average; see Fig. 6b) but with an opposite phase (Fig. 5 and 6a). The
amplitude of the $O_3$ response to VPSC reaches its maximum over the southern latitudes
during the spring (~60 DU; see Fig.7a for the austral spring period), which is consistent
with the role of PSCs on the polar $O_3$ depletion when there is sufficient sunlight. The
strong VPSC influence found at high northern latitudes in fall (Fig. 7a) are likely due to
compensation effects with EPF as pointed out above. Note also that the VPSC
contribution into MUSt reflects the larger correlation between the two stratospheric
layers over the southern polar region (Section 2.1, Fig. 1d).
5. AERO - Five important volcanic eruptions with stratospheric impact occurred during the
IASI mission (Kasatochi in 2008, Sarychev in 2009, Nabro in 2011, Sinabung in 2014
and Calbuco in 2015; see Fig.3). The two major eruptions of the last decades, El Chichon
(1982) and Mt. Pinatubo (1991), which have injected sulfur gases into the stratosphere,
have been shown to enhance PSCs particle abundances (~15-25 km altitude), to remove
$NO_x$ (through reaction with the surface of the sulfuric aerosol to form nitric acid) and,
hence, to make the ozone layer more sensitive to active chlorine (e.g. Hofmann et al.,
1989; Hofmann et al., 1993; Portmann et al., 1996; Solomon et al., 2016). Besides this
chemical effect, the volcanic aerosols also warm the stratosphere at lower latitudes
through scattering and absorption of solar radiation, which further induces indirect
dynamical effects (Dhomse et al., 2015; Revell et al., 2017). Even though the recent
eruptions have been of smaller magnitude than El Chichon and Mt. Pinatubo, they
produced sulphur ejection through the tropopause into the stratosphere (see Section 2.2,
Fig.2 and Fig.3), as seen with AOD reaching $5 \times 10^{-4}$ over the stratosphere (150-2 hPa),
especially following the eruptions of Nabro (13.3°N, 41.6°E), Sinabung (3.1°N, 98.3°E)
and Calbuco (41.3°S, 72.6°W). In the LSt, the regression supports an enhanced $O_3$
depletion over the Antarctic in presence of sulfur gases with a significantly negative
annual $O_3$ response reaching ~25 DU (i.e. relative contribution of ~20% into $O_3$ variation;
see Fig. 5b). On the contrary, enhanced $O_3$ levels in response to sulfuric acid are found in
the MUSt with a maximum impact of up to10 DU (i.e. relative contribution of ~20% into
the $O_3$ variation; see Fig. 5a) over the Antarctic. The change in phase in the $O_3$ response
to AERO between the LSt (~15-25 km) and the MUSt (~25-40 km) over the Antarctic, as





well as between polar and lower latitudes in the LSt (see Fig.5 and 6a), agree well with the heterogeneous reactions on sulfuric aerosol surface which reduce the concentration of $NO_x$ to form nitric acid, leading to enhanced $O_3$ levels above 25 km but leading to decreased $O_3$ levels due to chlorine activation below 25 km (e.g. Solomon et al., 1996). On a seasonal basis, the depletion due to the presence of sulfur gases reaches ~30 DU on a longitudinal average, over the S.H. polar region during the austral spring (see Fig.7a) highlighting the link between volcanic gases converted to sulfate aerosols and heterogeneous polar halogen chemistry.

6. NAO – The NAO is an important mode of global climate variability, particularly in northern winter. It describes large-scale anomalies in sea level pressure systems between the sub-tropical Atlantic (Azores; high pressure system) and sub-polar (Iceland; low pressure system) regions (Hurrell, 1995). It disturbs the location and intensity of the North Atlantic jet stream that separates these two regions depending on the phase of NAO. The positive (negative) phase of the NAO corresponds to larger (weaker) pressure difference between the two regions leading to stronger westerlies (easterlies) across the mid-latitudes (Barnston and Livezey, 1987). The two pressure system regions are clearly identified in the stratospheric $O_3$ response to NAO, particularly in the LSt, with positive regression coefficients above the Labrador-Greenland region and negative coefficients above the Euro-Atlantic region (Fig. 5b). Above these two sectors, the positive phase induces, respectively, an increase and a decrease in LSt $O_3$ levels. The negative phase is characterized by the opposite behaviour. That NAO pattern is in line with previous studies (Rieder et al., 2013) and was also observed from IASI in tropospheric $O_3$ (Wespes et al., 2017). The magnitude of annual LSt $O_3$ changes attributed to NAO variations reaches ~20 DU over the in-phase Labrador region (i.e. contribution of 25% relative to the $O_3$ variations), while a much lower contribution is found for the MUSt (~4 DU or ~10%). The NAO coeffficient in the LSt also shows that the influence of the NAO extends further into northern Asia in case of prolonged NAO phases. The NAO has also been shown to influence the propagation of waves into the stratosphere, hence, the BDC and the strength of the polar vortex in the N.H. mid-winter (Thompson and Wallace, 2000; Schnadt and Dameris, 2003; Rind et al., 2005). That connection between the NAO and the BDC might explain the negative anomaly in the $O_3$ response to EPF in the LSt over northern Asia which matches the region of negative response to the NAO.

7. AAO - The extra-tropical circulation of the S.H. is driven by the Antarctic oscillation that is characterized by geopotential height anomalies south of 20°S, with high anomalies of one sign centered in the polar region and weaker anomalies of the opposing sign north of 55°S (Thompson and Wallace, 2000). This corresponds well to the two band-like regions of opposite signs found for the regression coefficients of adjusted AAO in the LSt (negative coefficients centered in Antarctica and positive coefficient north of ~40°S; see





Fig.5b). Similarly to the N.H. mode, the strength of the residual mean circulation and of the polar vortex in the S.H. are modulated by the AAO through the atmospheric wave activity (Thompson and Wallace, 2000; Thompson and Salomon, 2001). During the positive (negative) phase of the AAO, the BDC is weaker (stronger) leading to less (more) $O_3$ transported from the tropics into the southern polar region, and the polar vortex is stronger (weaker) leading to more (less) $O_3$ depletion inside. This likely explains both the positive AAO coefficients in the region north of ~40°S (contribution < ~5 DU or ~10%) and the negative coefficients around and over the Antarctic (contribution reaching ~10 DU or ~15%; exception is found with positive coefficients over the western Antarctic). The dependence of $O_3$ variations to the AAO in the MUSt is lower than ~7 DU (or ~15%).

8. ENSO - Besides the NAO and the AAO, the El Nino southern oscillation is another dominant mode of global climate variability. This coupled ocean-atmosphere phenomenon is governed by sea surface temperature (SST) differences between high tropical and low extra-tropical Pacific regions (Harrison and Larkin, 1998). Domeisen et al. (2019) have recently reviewed the possible mechanisms connecting the ENSO to the stratosphere in the tropics and the extratropics of both hemispheres. The ozone response to ENSO is represented in Fig. 5 only for the ENSO-lag3 proxy which is found to be the main ENSO proxy contributing to the observed $O_3$ variations. While in the troposphere, previous works have shown that the ENSO influence mainly results in a high contrast of the regression coefficients between western Pacific/Indonesia/North Australia and central/eastern Pacific regions caused by reduced rainfalls and enhanced $O_3$ precursor emissions above western Pacific (called "chemical effect") (e.g. e.g. Oman et al., 2013; Valks et al., 2014; Ziemke et al., 2015; Wespes et al., 2016; and references therein), the LSt $O_3$ response to ENSO is shown here to translate into a strong tropical-extratropical gradient in the regression coefficients with a negative response in the tropics and a positive response at higher latitudes (~5 DU and ~10 DU, respectively, on longitudinal averages; see Fig. 6a). In the MUSt, ENSO is globally a smaller out-of-phase driver of $O_3$ variations (response of ~5 DU). The decrease in LSt $O_3$ during the warm ENSO phase in the tropics (characterized by a negative ENSO lag-3 coefficient reaching 7 DU (or 35%), respectively, in the LSt; see Fig. 5) is consistent with the ENSO-modulated upwelling via deep convection in the tropical lower stratosphere and, hence, increased BD circulation (e.g. Randel et al., 2009). The in-phase accumulation of LSt $O_3$ in the extra-tropics (contribution reaching 15 DU or 20%; see Fig. 5) is also consistent with enhanced extra-tropical planetary waves that propagate into the stratosphere during the warm ENSO phase, resulting in sudden stratospheric warmings and, hence, in enhanced BDC and weaker polar vortices (e.g. Brönnimann et al., 2004; Manzini et al., 2006; Cagnazzo et al., 2009). The very pronounced link between stratospheric $O_3$ and the ENSO related dynamical pathways with a time lag of about 3 months is one key finding of the present



work. Indeed, if the influence of ENSO on stratospheric O$_3$ measurements has been reported in earlier studies (Randel and Cobb, 1994; Brönnimann et al., 2004; Randel et al., 2009; Randel and Thompson, 2011; Oman et al., 2013; Manatsa and Mukwada, 2017; Tweedy et al., 2018), it is the first time that a delayed stratospheric O$_3$ response is investigated in MLR studies. A 4- to 6-month time lag in O$_3$ response to ENSO has similarly been identified from IASI in the troposphere (Wespes et al., 2017), where it was explained not only by a tropospheric pathway but also by a specific stratospheric pathway similar to that modulating stratospheric O$_3$ but with further impact downward onto tropospheric circulation (Butler et al., 2014; Domeisen et al., 2019). Furthermore, the 3-month lag identified in the LSt O$_3$ response is fully consistent with the modelling work of Cagnazzo et al. (2009) that reports a warming of the polar vortex in February-March following a strong ENSO event (peak activity in November-December) associated with positive O$_3$ ENSO anomaly reaching ~10 DU in the Arctic and negative anomaly of ~6-7 DU in the Tropics. We find that the tropical-extra-tropical gradient in O$_3$ response to ENSO-lag3 is indeed much stronger in spring with contributions of ~20-30 DU (see Fig.7a for the austral spring period *vs* winter).

Overall, although the annual MLR model underestimates the O$_3$ variability at high latitudes (>50°N/S) by up to 5 DU, particularly in the MUSt (see Fig. 6b), we conclude that it gives a good overall representation of the sources of O$_3$ variability in the two stratospheric layers sounded by IASI. This is particularly true for the spring period (see Fig. 7) which was studied in several earlier works to reveal the onset of Antarctic total O$_3$ recovery (Salby et al., 2011; Kuttippurath et al., 2013; Shepherd et al., 2014; Solomon et al., 2016; Weber et al., 2018), despite the large O$_3$ variability due to the hole formation during that period (~80 DU). It is also interesting to see from Fig.7 that the broad O$_3$ depletion over Antarctica in the LSt is attributed by the MLR to VPSC (up to 60 DU of explained O$_3$ variability on a latitudinal average). Following these promising results, we further analyze below the O$_3$ variability in response to anthropogenic perturbations, assumed in the MLR model by the linear trend term, with a focus over the polar regions.

## 4 Trend analysis

### 4.1 10-year trend detection in stratospheric layers

The distributions of the linear trend estimated by the annual regression are represented in Fig. 8a for the MUSt and the LSt (left and right panels). In agreement with the early signs of O$_3$ recovery reported for the extra-tropical mid- and upper stratosphere above ~25-10 hPa (>25-30 km; Pawson et al., 2014; Harris et al., 2015; Steinbrecht et al., 2017; Sofieva et al., 2017; Ball et al., 2018), the MUSt shows significant positive trends larger than 1 DU/yr poleward of ~35°N/S. The corresponding decadal trends (>10 DU/dec) are much larger than the discontinuity of ~2-4



DU encountered in the MUSt record on 15 September 2010 and discussed in section 2.1. The tropical MUSt also shows positive trends but they are weaker (<0.8 DU/yr) or not significant. The largest increase is observed in polar $O_3$ with amplitudes reaching ~2.5 DU/yr. The mid-latitudes also show significant $O_3$ enhancement which can be attributed to airmass mixing after the disruption of the polar vortex (Knudsen and Grooss, 2000; Fioletov and Shepherd, 2005; Dhomse, 2006; Nair et al., 2015).

As in the MUSt, the LSt is characterized in the southern polar latitudes by significantly positive and large trends (between ~ |1.0| and |2.5| DU/yr). In the mid-latitudes, the lower stratospheric trends are significantly negative, i.e. opposite to those obtained in the MUSt. This highlights the independence between the two $O_3$ layers sounded by IASI in the stratosphere. Poleward of 25°N the negative LSt trends range between ~ |0.5| and |2.0| DU/yr. Negative trends in lower stratospheric $O_3$ have already been reported in extra-polar regions from other space-based measurements (Kyrölä et al., 2013; Gebhardt et al., 2014; Sioris et al., 2014; Harris et al., 2015; Nair et al., 2015; Vigouroux et al., 2015; Wespes et al., 2016; Steinbrecht et al., 2017; Ball et al., 2018) and may be due to changes in stratospheric dynamics at the decadal timescale (Galytska et al., 2019). These previous studies, which were characterized by large uncertainties or resulted from composite-data merging techniques, are confirmed here using a single dataset. The negative trends which are observed at lower stratospheric middle latitudes are difficult to explain with chemistry-climate models (Ball et al., 2018). It is also worth noting that the significant MUSt and LSt $O_3$ trends are of the same order as those previously estimated from IASI over a shorter period (from 2008 to 2013) and latitudinal averages (see Wespes et al., 2016). This suggests that the trends are not very sensitive to the natural variability in the IASI time series, hence, supporting the significance of the $O_3$ trends presented here.

The sensitivity of IASI $O_3$ to the estimated trend from MLR is further verified in Fig. 8b that represents the global distributions of relative differences in the *RMSE* of the regression residuals obtained with and without a linear trend term included in the MLR model (($RMSE_{w/o\_LT}$ − $RMSE_{with\_LT}$)/$RMSE_{with\_LT}$ ×$100$; in %). An increase of 1.0-4.0% and 0.5-2.0% in the *RMSE* is indeed observed for both the MUSt and the LSt, respectively, in regions of significant trend contribution (Fig. 8a), when the trend is excluded. This demonstrates the significance of the trend in improving the performance of the regression. Another statistical method that can be used for evaluating the possibility to infer, from the IASI time period, the significant positive or negative trends in the MUSt and the LSt, respectively, consists in determining the expected year when these specified trends would be detectable from the available measurements (with a probability of 90%) by taking into account the variance ($\sigma_{\varepsilon}^2$) and the autocorrelation ($\Phi$) of the noise residual according to the formalism of Tiao et al. (1990) and Weatherhead et al. (1998). It represents a more drastic and conservative method than the standard MLR. The results are displayed in Fig. 8c for an assumed specified trend of |1.5| DU/yr, which corresponds to a





medium amplitude of trends derived here above from the MLR over the mid-polar regions (Fig.
8a). In the MUSt, we find that ~2-3 additional years of IASI measurements would be required to
unequivocally detect a positive trend of |1.5| DU/yr (with probability 0.90) over high latitudes
(detectable from ~2020-2022 ± 6-12 months) whereas it should already be detectable over the
mid- and lower latitudes (from ~2015 ± 3-6 months). In the LSt, additional ~7 years (± 1-2
years) of IASI measurement would be required to categorically identify the probable decline
derived from the MLR in northern mid-latitudes, and even more to measure the enhancement in
the southern polar latitudes. The longest required measurement period over the high latitudes is
explained by the largest noise residual (i.e. largest $\sigma_\varepsilon$) in the IASI data (see Fig.4 a and b). Note
that a larger specified trend amplitude would obviously require a shorter period of IASI
measurement. Only ~2 additional years would be required to detect a specified trend of |2.5|
DU/yr which characterizes the LSt at mid-high latitudes.
**4.2 Stratospheric contributions to total O$_3$ trend**
The effect on total O$_3$ of the counteracting trends in the northern mid-latitudes and of the
constructive trends in the southern polar latitudes trends derived in the two stratospheric layers
sounded by IASI is now investigated.
Figure 9 represents the global distributions of the contribution of the MUSt and the LSt into the
total O$_3$ columns (Fig.9a; in %), of the adjusted trends for the total O$_3$ (Fig. 9b in DU/yr) and of
the estimated year for a |15| DU per decade trend detection with a probability of 90% (Fig. 9c).
While no significant change or slightly positive trends in total O$_3$ after the inflection point in
1997 have been reported on an annual basis (e.g; Weber et al., 2018), Fig. 9b shows clear
significant changes: negative trend at northern mid-latitudes (up to ~2.0 DU/yr north of 30°N)
and positive trend over the southern polar region (up to ~3.0 DU/yr south of 40°S). Although
counteracting trends between lower and upper stratospheric O$_3$ have been pointed out in the
recent study of Ball et al. (2018) to explain the non-significant recovery in total O$_3$, we find from
IASI a dominance of the LSt decline that translates to negative trends over some regions of the
N.H. in TOC (Fig. 9b). This is explained by the contributions of 45-55% from the LSt to the total
column, *vs* ~30-40% from the MUSt (Fig. 9a) in the mid- and polar regions over the whole year.
In addition, the significant positive trends over the high southern latitudes in both the MUSt and
the LSt explains the largest total O$_3$ enhancement in polar region. Note that most previous ozone
trends studies, including Ball et al. (2018), excluded the polar regions due to limited latitude
coverage of some instruments merged in the data composites.
While the annual MLR shows a significant dominance of LSt trends over MUSt trends in the
northern mid-latitudes and significant constructive trends in the southern latitudes, total O$_3$
trends are not ascribed with complete confidence according to the formalism of Tiao et al. (1990)



and Weatherhead et al. (1998) discussed in Section 4.1. The detectability of a specified trend of
|1.5| DU/yr (Fig. 9c), which corresponds to the medium trend derived from MLR in mid-high
latitudes of both hemispheres (Fig. 9b), would need several years of additional measurements to
be unequivocal from IASI on an annual basis (from ~2022-2024 over the mid-latitudes and from
~2035 over the polar regions). The highest trend amplitude of ~|2.5| DU/yr derived from the
MLR would be observable from ~2020-2025 (figure not shown).
The use of the annual MLR could translate to large systematic uncertainties on trends (implying
large $\sigma_\varepsilon$ ), which induces a longer measurement period required to yield significant trends. These
uncertainties could be reduced on a seasonal basis, by attributing different weights to the
seasons, which would help in the categorical detection of a specified trend. This is investigated
in the subsection below by focusing on the winter and the spring periods.

### 4.3 Trends in polar spring

The reports on early signs of total $O_3$ recovery (Salby et al., 2011; Kuttippurath et al., 2013;
Shepherd et al., 2014; Solomon et al., 2016; Kuttippurath and Nair, 2017; Weber et al., 2018)
have all focused on the Antarctic region during spring, when the ozone hole area is at its
maximum extent, i.e. the LSt $O_3$ levels at minimum values. Here we investigate the respective
contributions of the LSt and the MUSt to the TOC recovery over the South Pole, looking also at
the JJA period because the minima in $O_3$ levels in the MUSt over Antarctica occur later in
summer (down to ~80 DU; see Fig.4a). Figures 10 and 11, respectively, show the S.H. and the
N.H. distribution of the estimated trends from seasonal MLR (left panels) and of the
corresponding year required for a significant detection of |30| DU increase per decade (right
panels) during their respective winter (JJA and DJF; Fig. 10a and 11a) and spring (SON and
MAM; Fig. 10b and 11b) for the total, MUSt and LSt $O_3$ (top, middle and bottom panels,
respectively). Fig. 10 a and b clearly show significant positive trends over Antarctica and the
southernmost latitudes of the Atlantic and Indian oceans, with amplitudes ranging between ~1-5
DU/yr over latitudes south of ~35-40°S in total, MUSt and LSt $O_3$ (~3.9±1.7 DU/yr, ~2.7±1.0
DU/yr, ~3.3±2.6 DU/yr and ~4.4±1.9 DU/yr, ~1.6±0.6 DU/yr, ~3.4±1.4 DU/yr, on spatial
averages, respectively over JJA and SON, for the three $O_3$ columns). These trends are much
larger than the amplitude of the discontinuity in the MUSt time series (section 2.1) and than the
annual ones estimated in Sections 4.1 (see Fig.8 for the MUSt and the LSt) and 4.2 (see Fig.9 for
TOC) over the whole year. In MUSt, significant positive trends are observed during each season
over the mid- and polar latitudes of both hemispheres (Fig. 10 and 11 for the winter and spring
periods; the other seasons are not shown here) but more particularly in winter and in spring
where the increase reaches a maximum of ~ 4 DU/yr. In the LSt, the distributions are more
complex: the trends are significantly negative in the mid-latitudes of both hemispheres,
especially in winter, and in spring of the N.H., while in spring of the S.H., some mid-latitude



regions also show near-zero or even positive trends. The southern polar region shows high
significant positive trends in winter/spring (see Fig.10). For the total $O_3$ at mid-high latitudes,
given the mostly counteracting trends detected in the LSt and in the MUSt and the dominance of
the LSt over the MUSt (~45-55% from the LSt vs ~30-40% from the MUSt into total $O_3$ over the
whole year; except over the Antarctica in spring as discussed above), these latitudes are
governed by negative trends with the highest decline in spring of the N.H. High significant
increases are detected over polar regions in winter/spring of both hemispheres but more
particularly in the S.H. where the LSt and MUSt trends are both of positive sign.

The substantial winter/spring positive trends observed in MUSt, LSt and total $O_3$ levels at high
latitudes of the S.H. (and of the N.H. for the MUSt) are furthermore demonstrated to be
detectable from the available IASI measurement period (see Fig. 10, right panels: an assumed
increase of |3| DU/yr is detectable from 2016 ± 6 months and from 2018 ± 1 year in the MUSt
and the LSt, respectively). The positive trend of ~4 DU/yr measured in polar total $O_3$ in
winter/spring would be observable from ~2018-2020 ± 1-2 year and the decline of ~|3| DU/yr in
winter/spring of the N.H. in LSt would be detectable from ~2018-2020 ± 9 months (not shown
here). Note that the unrealistic negative trends found above the Pacific at highest latitudes (see
Fig. 10) correspond to the regions with longest required measurement period for trend detection
and, hence, point to poor regression residuals. About ~50% and ~35% of the springtime MUSt
and LSt $O_3$ variations, respectively, are due to anthropogenic factors (estimated by VPSC×EESC
proxy and linear trend in MLR models). This suggests that $O_3$ changes especially in the LSt are
mainly governed by dynamics, which contributes to a later trend-detection year in comparison
with the MUSt (Fig. 10 and 11) and which may hinder the $O_3$ recovery process.

Overall, the large positive trends estimated concurrently in LSt, MUSt and total $O_3$ over the
Antarctic region in winter/spring likely reflect the healing of the ozone layer with a decrease of
polar ozone depletion (Salomon et al., 2016) and, hence, demonstrate the efficiency of the
Montreal protocol. To the best of our knowledge, these results represent the first detection of a
significant recovery in the stratospheric and the total $O_3$ columns over the Antarctic from one
single satellite dataset.

**4.4 Speeding up in $O_3$ changes**

Positive trends in total $O_3$ have already been determined earlier by Solomon et al. (2016) and by
Weber et al. (2018) over Antarctica during September over earlier periods (~2.5±1.5DU/yr over
2000-2014 and 8.2±6.2%/dec over 2000-2016, respectively). The larger trends derived from the
IASI records (Fig.10b; ~3.9±1.7 DU/yr on average) suggest that the $O_3$ response could be
speeding up due to the accelerating decline of $O_3$ depleting substances (ODS) resulting from the
Montreal Protocol. This has been investigated here by estimating the change in trend in MUSt,
LSt and total $O_3$ over the IASI mission. Knowing that the length of the measurement period is an



important criterion for reducing systematic errors in the trend coefficient determination (i.e. the specific length of natural mode cycles should be covered to avoid any possible compensation effect between the covariates), the ozone response to each natural driver is taken from their adjustment over the whole IASI period (2008-2017; Section 3, Fig.5) and kept fixed. The linear trend term only is adjusted over variable measurement periods that all end in December 2017, by using a single linear iteratively reweighted least squares regression. The results are displayed in Fig. 12 for total, MUSt and LSt $O_3$ trends and their associated uncertainty (in the 95% confidence level) estimated from an annual regression. Note that the results are only shown for periods starting before 2015 as too short periods induce too large standard errors. In the LSt, a clear speeding up in the southern polar $O_3$ recovery is observed with amplitude ranging from ~1.5±0.3 DU/yr over 2008-2017 to ~6.5±3.5 DU/yr over 2015-2017 on latitudinal averages. Similarly, a speeding of the $O_3$ decline at northern mid-latitudes is found with values ranging between ~-0.7±0.2 DU/yr over 2008-2017 and ~-2.5±1.5 DU/yr over 2015-2017. In the MUSt, a weaker increase is observed over the year around ~60° latitude of the S.H. (from ~1.0±0.1 DU/yr over 2008-2017 to ~3.5±2.0 DU/yr over 2015-2017). Given the positive acceleration in both LSt and MUSt $O_3$ in the S.H., this is where the total $O_3$ record is characterized by the largest significant recovery (from ~1.5±0.3 DU/yr over 2008-2017 to ~8.5±3.5 DU/yr over 2015-2017). Surprisingly, the speeding up in the $O_3$ decline in the N.H. is more pronounced in the total $O_3$ (from ~-1.0±0.2 DU/yr over 2008-2017 to ~-3.5±1.5 DU/yr over 2015-2017) compared to the LSt, despite the opposite trend in MUSt $O_3$. This could reflect the $O_3$ decline observed in the northern latitudes in the troposphere (~|0.5| DU/yr over 2008-2016; cfr Wespes et al., 2018) which is included in the total column.

Overall, the larger annual trend amplitudes derived over the last few years of total, MUSt and LSt $O_3$ measurements, compared with those derived from the whole studied period (Sections 4.1 and 4.2) and from earlier studies translate to trends that are categorically detectable over the covered period. This demonstrates that we progress towards a significant emergence and speeding up of $O_3$ recovery process in the stratosphere over the whole year.

**5 Summary and conclusion**

In this study, we have analysed the changes in stratospheric $O_3$ levels sounded by IASI-A by examining the global pictures of natural and anthropogenic sources of $O_3$ changes independently in the lower (150-25 hPa) and in the mid-upper stratosphere (<25 hPa). We have exploited to that end a multi-linear regression model that has been specifically developed for the analysis of stratospheric processes by including a series of drivers known to have a causal relationship to natural stratospheric $O_3$ variations, namely SF, QBO-10, QBO-30, NAO, AAO, ENSO, AERO, EPF and VPSC. We have first verified the representativeness of the $O_3$ response to each of these natural drivers and found for most of them characteristic patterns that are in line with the current knowledge of their dynamical influence on $O_3$ variations. One of the most important finding



related to the O₃ driver analysis relied on the detection of a very clear time lag of 3 months in the O₃ response to ENSO in the LSt, with a pronounced contrast between an in-phase response in the extra-tropics and an out-of-phase response in the tropics, which is consistent with the ENSO-modulated dynamic. The 3-month lag observed in the lower stratosphere is also coherent with the 4- to -6 months lag detected from a previous study in the troposphere (Wespes et al., 2017) and further supports the stratospheric pathway suggested in Butler et al. (2014) to explain an ENSO influence over a long distance. The representativeness of the influence of the O₃ drivers was also confirmed on a seasonal basis (e.g. high ENSO-lag3 effect in spring, strong VPSC and AERO influences during the austral spring …). These results have verified the performance of the regression models (annual and seasonal) to properly discriminate between natural and anthropogenic drivers of O₃ changes. The anthropogenic influence has been evaluated with the linear trend adjustment in the MLR. The main results are summarized as follows:

(i) A highly probable (within 95%) recovery process is derived from the annual MLR at high southern latitudes in the two stratospheric layers and, therefore, in the total column. It is also derived at high northern latitudes in the MUSt. However, the effectiveness of the Montreal Protocol needs a longer period of IASI measurements for being unequivocally assured. Only ~2-3 additional years of IASI measurements are required in the MUSt.

(ii) A likely O₃ decline (within 95%) is measured in the lower stratosphere at mid-latitudes, specifically, of the N.H., but it would require an additional ~7 years of IASI measurements to be categorically confirmed. Given the large contribution from the LSt to the total column (~45-50% from LSt vs ~35% from the MUSt into TOCs), the decline is also calculated in total O₃ with ~4-6 years of additional measurements for being unequivocal.

(iii) A significant O₃ recovery is categorically found in the two stratospheric layers (>~35°N/S in the MUSt and >~45°S in the LSt) as well as in the total column (>~45°S) during the winter/spring period, which confirms previous studies that showed healing in the Antarctic O₃ hole with a decrease of its areal extent. These results verify the efficiency of the Montreal protocol with the banning of ODS, through the stratosphere and in the total column, from only one single satellite dataset for the first time.

(iv) The decline observed in LSt O₃ at northern mid-latitudes is unequivocal over the available IASI measurements in winter/spring of the N.H. The exact reasons for that decline are still unknown but O₃ changes in the LSt are estimated to be mainly attributable to dynamics and it likely perturbs the healing of LSt and total O₃ in the N.H.



(v) A significant speeding up (within 95%) in that decline is measured in LSt and total $O_3$
over the last 10 years (from ~-0.7±0.2 DU/yr over 2008-2017 to ~-2.5±1.5 DU/yr
over 2015-2017 in LSt $O_3$ on latitudinal averages). It is of particular urgency to
understand its causes for apprehending its possible impact on the $O_3$ layer and on
future climate changes.

(vi) A clear and significant speeding up (within 95%) in stratospheric and total $O_3$ recovery
is measured at southern latitudes (e.g. from ~1.5±0.3 DU/yr over 2008-2017 to
~6.5±3.5 DU/yr over 2015-2017 in the LSt) and translate to trend values that are
categorically detectable on an annual basis. It demonstrates that we are currently
progressing towards a substantial emergence in $O_3$ healing in the stratosphere over the
whole year in the S.H..


Additional years of IASI measurements that will be provided by the operational IASI-C (2018)
on flight and the upcoming IASI-Next Generation (IASI-NG) instrument onboard the Metop
Second Generation (Metop-SG) series of satellites would be of particular interest to confirm and
monitor, in a near future and over a longer period, the speeding up in the $O_3$ healing of the S.H.
as well as in the LSt $O_3$ decline measured at mid-latitudes of the N.H. IASI-NG/Metop-SG is
expected to extent the data record much further in the future (Clerbaux and Crevoisier, 2013;
Crevoisier et al., 2014).

**Author contribution**

C.W. performed the analysis, wrote the manuscript and prepared the figures. D.H. was
responsible for the retrieval algorithm development and the processing of the IASI $O_3$ dataset.
S.C. and P.-F.C. contributed to the analysis. All authors contributed to the interpretation of the
results and reviewed the manuscript.

**Acknowledgments**

IASI has been developed and built under the responsibility of the Centre National d'Etudes
Spatiales (CNES, France). It is flown onboard the Metop satellites as part of the EUMETSAT
Polar System. The IASI L1 data are received through the EUMETCast near real time data
distribution service. We acknowledge the financial support from the ESA $O_3$-CCI and
Copernicus $O_3$-C3S projects. FORLI-$O_3$ is being implemented at Eumetsat with the support of
the AC SAF project. The research in Belgium is also funded by the Belgian State Federal Office
for Scientific, Technical and Cultural Affairs and the European Space Agency (ESA Prodex IASI
Flow and B-AC SAF). We acknowledge Ingo Wohltmann (Alfred Wagner Institute, Postdam,
Germany) as well as Beiping Luo (Institute for Atmosphere and Climate, ETH Zurich,



Switzerland) and Larry Thomason (NASA Langley Research Center, Hampton, USA), for
processing and providing datasets of volume of polar stratospheric clouds and of sulfuric acid
extinction coefficients, respectively. We are also grateful to Maxime Prignon (Université de
Liège, Liège, Belgium) for providing several years of BASCOE simulations.





**Table 1** List of the explanatory variables used in the multi-linear regression model applied on
IASI stratospheric $O_3$, their temporal resolution and their sources.

| Proxy | Description (*resolution*) | Sources |
|---|---|---|
| **F10.7** | The 10.7 cm solar radio flux (*daily*) | NOAA National Weather Service Climate Prediction Center: ftp://ftp.ngdc.noaa.gov/STP/space-weather/solar-data/solar-features/solar-radio/noontime-flux/penticton/penticton_adjusted/listings/listing_drao_noontime-flux-adjusted_daily.txt |
| **QBO10** **QBO30** | Quasi-Biennial Oscillation index at 10hPa and 30hPa (*monthly*) | Free University of Berlin: www.geo.fu-berlin.de/en/met/ag/strat/produkte/qbo/ |
| **EPF** | Vertical component of Eliassen-Palm flux crossing 100 hPa, averaged over 45°-75° for each hemisphere and accumulated over the 3 or 12 last months (see text for more details) (*daily*) | Calculated at ULB from the NCEP/NCAR gridded reanalysis: https://www.esrl.noaa.gov/psd/data/gridded/data.ncep.reanalysis.html |
| **AERO** | Stratospheric volcanic aerosols; Vertically integrated sulfuric acid extinction coefficient at 12 µm over 150-25 hPa and 25-2hPa, averaged over the tropics and the extra-tropics north and south (see text for more details) (*monthly*) | Extinction coefficients processed at the Institute for Atmosphere and Climate (ETH Zurich, Switzerland; Thomason et al., 2018) |
| **VPSC** | Volume of Polar Stratospheric Clouds for the N.H. and the S.H. multiplied by the equivalent effective stratospheric chlorine (EESC) and accumulated over the 3 or 12 last months (see text for details) (*daily*) | Processed at the Alfred Wagner Institute (AWI, Postdam, Germany; Ingo Wolthmann, private communication)  EESC taken from the Goddard Space Flight Center: https://acd-ext.gsfc.nasa.gov/Data_services/automailer/index.html |
| **ENSO** | Multivariate El Niño Southern Oscillation Index (MEI) (2-*monthly averages*) | NOAA National Weather Service Climate Prediction Center: http://www.esrl.noaa.gov/psd/enso/mei/table.html |
| **NAO** | North Atlantic Oscillation index for the N.H. (*daily*) | ftp://ftp.cpc.ncep.noaa.gov/cwlinks/norm.daily.nao.index.b500101.current.ascii |
| **AAO** | Antarctic Oscillation index for the S.H. (*daily*) | ftp://ftp.cpc.ncep.noaa.gov/cwlinks/norm.daily.aao.index.b790101.current.ascii |
| **GEO** | Geopotential height at 200 hPa (*daily*) | http://apps.ecmwf.int/datasets/data/interim-full-daily/?levtype=pl |
| **PV** | Potential vorticity at 200 hPa (*daily*) | |




**Figure captions**


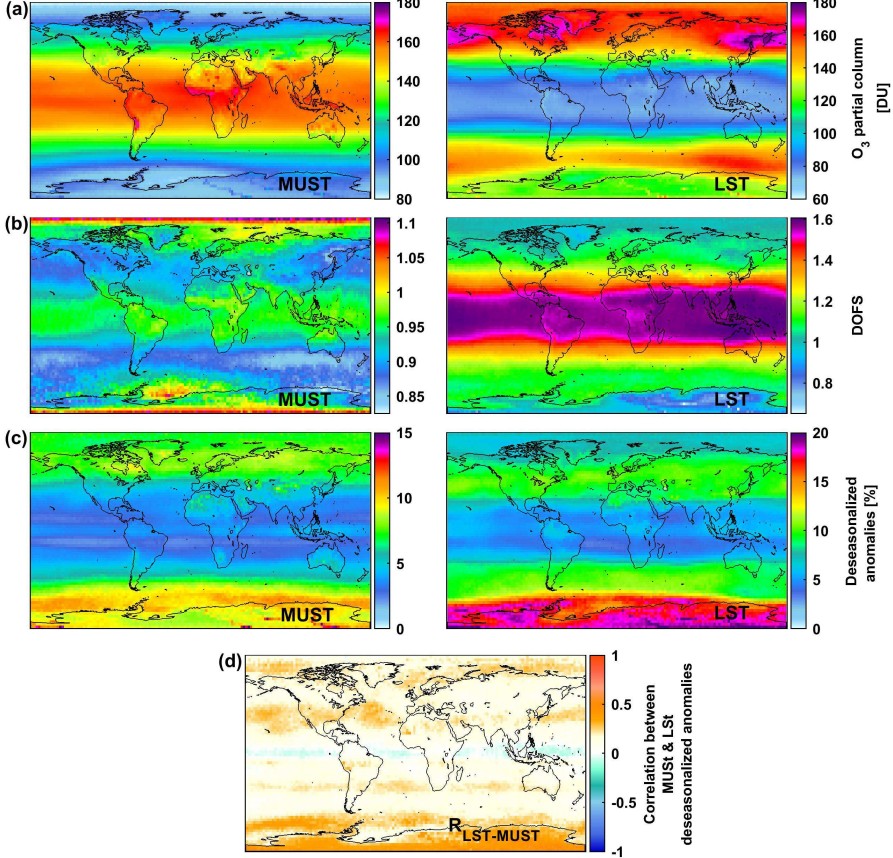


**Fig.1.** Global distribution of (a) daily $O_3$ columns (in Dobson Units - DU), (b) associated DOFS,
(c) absolute deseasonalized anomalies (in %) averaged over January 2008 – December 2017 in
the MUSt (Mid-Upper Stratosphere: >25 hPa; left panels) and in the LSt (Lower Stratosphere:
150-25hPa; right panels). (d) shows the correlation coefficients between the daily $O_3$
deseasonalized anomalies in the MUSt and in the LSt. Note that the scales are different between
MUSt and LSt.






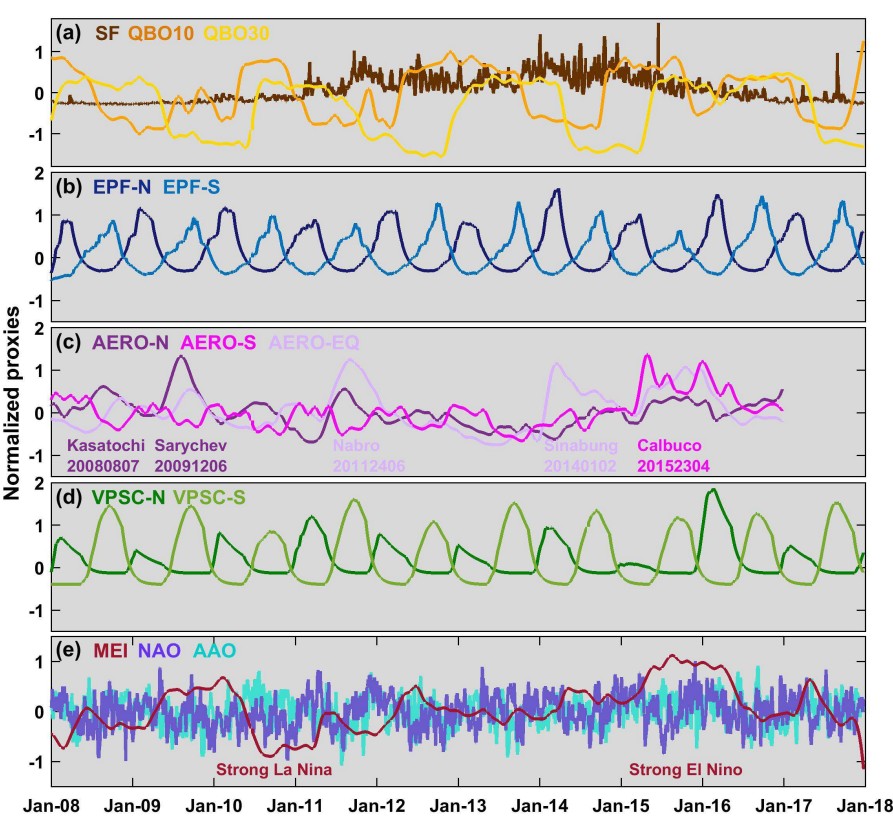


**Fig.2.** Normalized proxies as a function of time for the period covering January 2008 to December 2017 for (a) the F10.7 cm solar radio flux (SF) and the equatorial winds at 10 (QBO10) and 30 hPa (QBO30), respectively, (b) the upward components of the EP flux crossing 100 hPa accumulated over time and averaged over the 45°-75° latitude band for each hemispheres (EPF-N and EPF-S), (c) the extinction coefficients at 12 µm vertically integrated over the stratospheric $O_3$ column (from 150-2hPa) and averaged over the extra-tropics north and south (22.5°-90°N/S; AERO-N and AERO-S) and over the tropics (22.5°S-22.5°N; AERO-EQ) (the main volcanic eruptions are indicated), (d) the volume of polar stratospheric clouds multiplied by the equivalent effective stratospheric chlorine (EESC) and accumulated over time for the north and south hemispheres (VPSC-N and VPSC-S) and (e) the El Niño Southern (ENSO), North Atlantic (NAO) and Antarctic (AAO) oscillations.

900
901





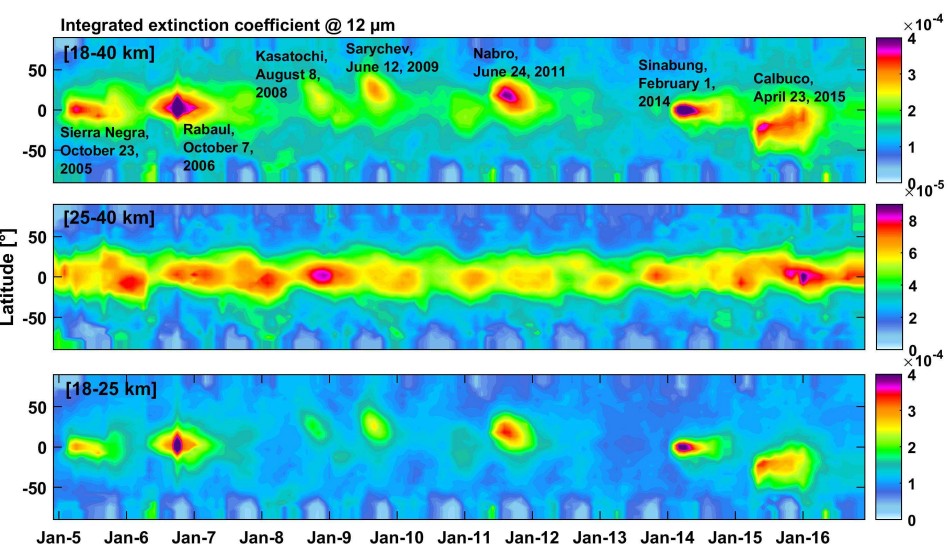

**Fig.3:** Latitudinal distribution of volcanic sulfuric acid extinction coefficient at 12 µm integrated over the stratosphere (top panel), over the middle stratosphere (middle panel) and the lower stratosphere (bottom panel) as a function of time from 2005 to 2017. The dataset consists of monthly mean aerosol data merged from SAGE, SAM, CALIPSO, OSIRIS, 2D-model-simulation and Photometer (processed at NASA Langley Research Center, USA and ETH Zurich, Switzerland).

927

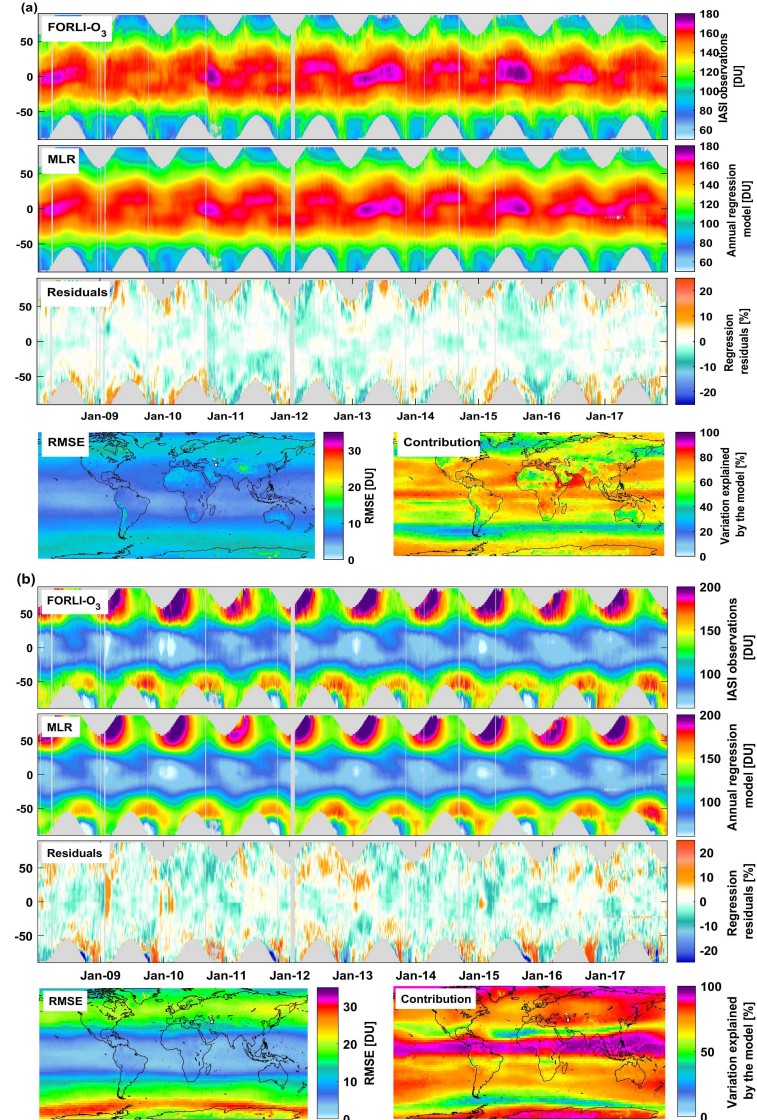

928
929
**Fig.4:** Latitudinal distribution of (a) MUSt $O_3$ column and (b) LSt $O_3$ columns as a function of
time observed from IASI (in DU; top panels), simulated by the annual regression model (in DU,
second panels) and of the regression residuals (in DU; third panels). Global distribution of *RMSE*
of the regression residual (in DU) and fraction of the variation in IASI data explained by the
regression model calculated as $\left[100 \times \left(\sigma\left(O_3^{Fitted-model}(t)\right) \middle/ \sigma\left(O_3(t)\right)\right)\right]$ (in %; fourth panels).

935



936

937

**Fig.5:** Global distribution of the annual regression coefficient estimates (in DU) for the main O$_3$ drivers in (a) MUSt and in (b) LSt: QBO10, QBO30, SF, EPF, VPSC, AERO, NAO, AAO and ENSO (ENSO-lag3 for both LSt and MUSt). Grey areas and crosses refer to non-significant grid cells in the 95% confidence limit. Note that the scales differ among the drivers.





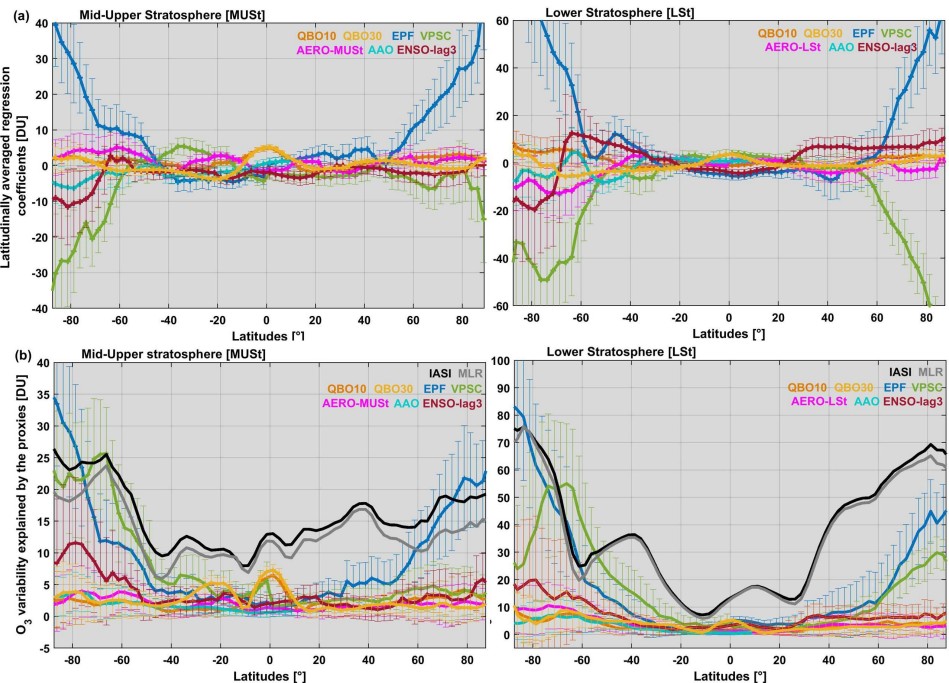

**Fig.6**: Latitudinal distributions (a) of fitting regression coefficients for various $O_3$ drivers
(QBO10, QBO30, EPF, VPSC, AERO, AAO and ENSO-lag3; in DU) and (b) of 2σ $O_3$
variability due to variations in those drivers (in DU) from the annual MLR in MUSt and LSt (left
and right panels respectively). Vertical bars correspond (a) to the uncertainty of fitting
coefficients at the 2σ level and (b) to the corresponding error contribution into $O_3$ variation. Note
that the scales are different.





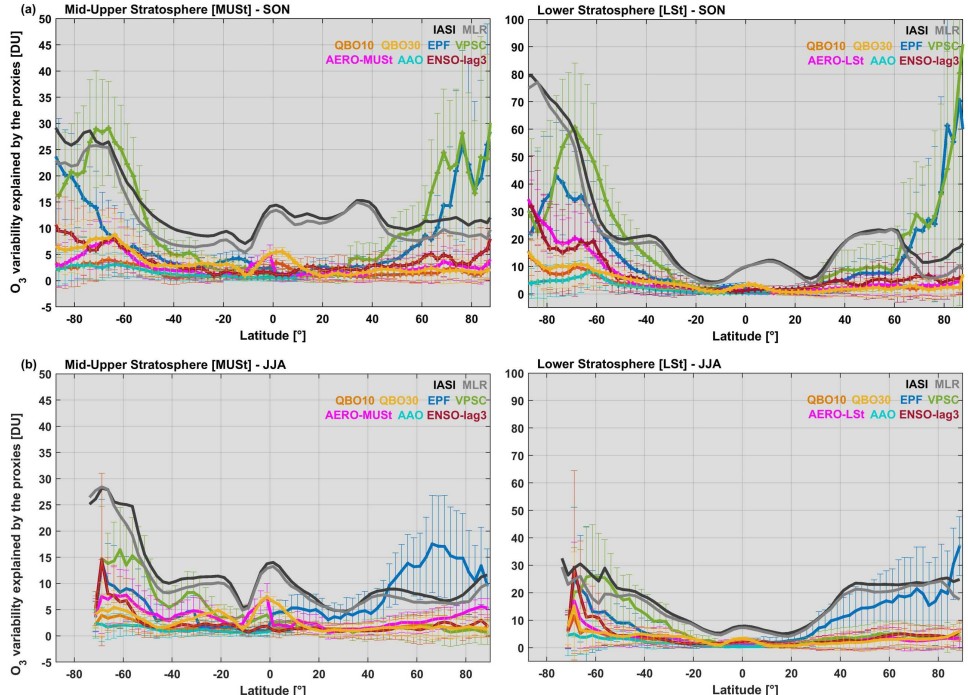

**Fig.7**: Same as Fig. 6b but for (a) the austral winter and (b) the austral spring periods (JJA and SON, respectively) from the seasonal MLR. Note that the scales are different.





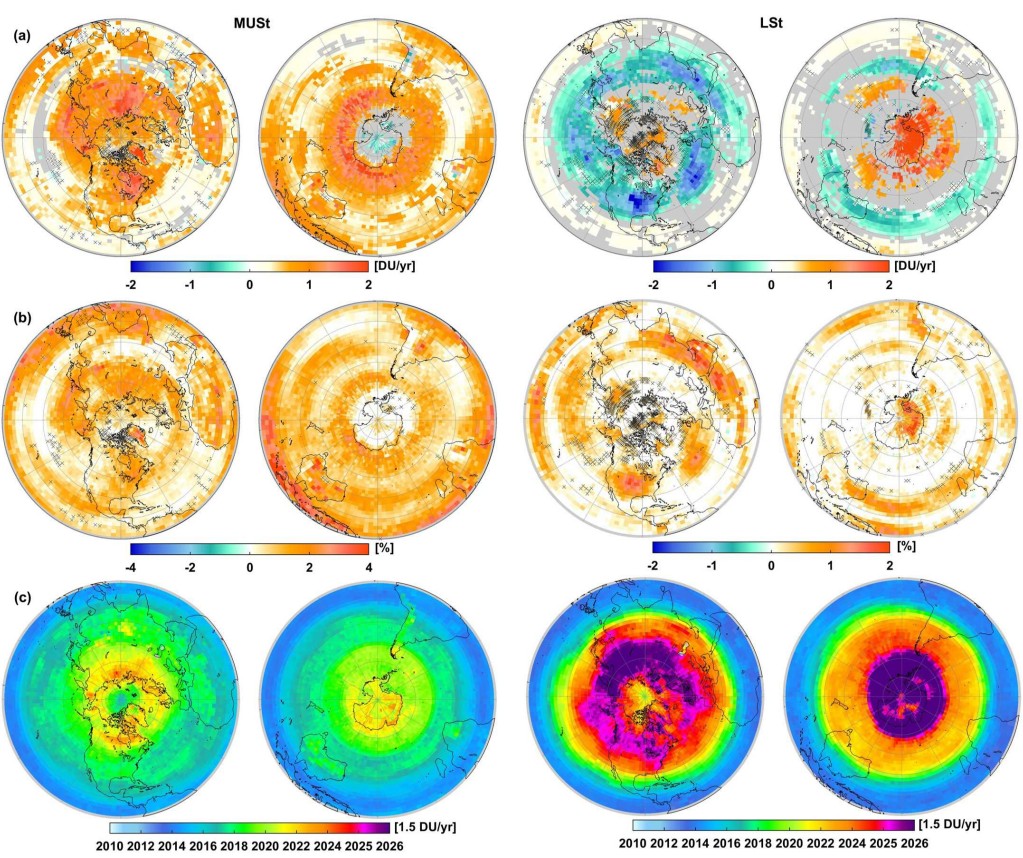

**Fig. 8:** Global distribution (a) of the estimated annual trends (in DU/yr; grey areas and crosses refer to non-significant grid cells in the 95% confidence limit), (b) of the IASI sensitivity to trend calculated as the differences between the *RMSE* of the annual MLR fits with and without linear trend term [(*RMSE$_{w/o\_LT}$* − *RMSE$_{with\_LT}$*)/*RMSE$_{with\_LT}$* ×*100*] (in %), (c) of the estimated year for a significant detection (with a probability of 90%) of a given trend of |1.5| DU/yr starting in January 2008 in MUSt and LSt O$_3$ columns (left and right panels, respectively). Note that the scales of panels (b) are different for the two layers.



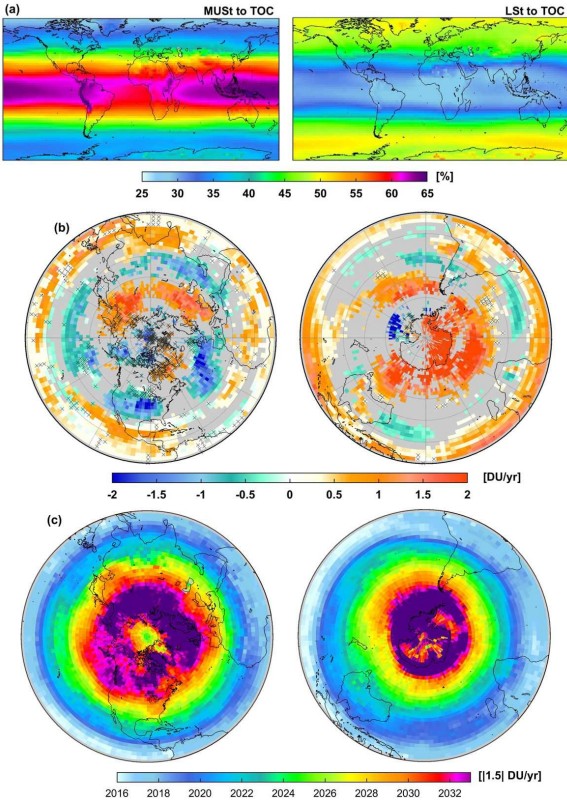

970
971

**Fig.9:** Global distribution of (a) the contribution (in %) of MUSt and LSt into the total $O_3$ (left and right panels respectively) averaged over January 2008 – December 2017, (b) fitted trends in total $O_3$ (in DU/yr; the grey areas and crosses refer to the non-significant grid cells in the 95% confidence limit) and (c) estimated year for the detection of a significant trend in total $O_3$ (with a probability of 90%) for a given trend of |1.5| DU/yr starting on January 2008.





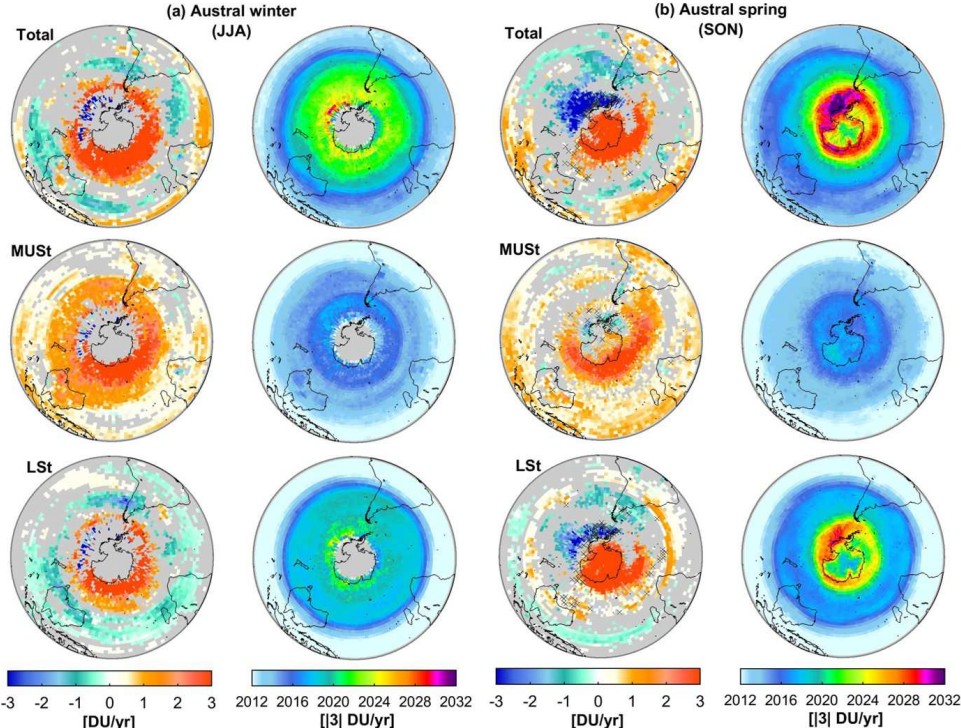

**Fig.10:** Hemispheric distribution (a) in austral winter (JJA) and (b) in austral spring (SON) of the estimated trends in total, MUSt and LSt $O_3$ columns (left panels: top, middle and bottom, respectively; in DU/yr; the grey areas and crosses refer to the non-significant grid cells in the 95% confidence limits) and of the corresponding estimated year for a significant trend detection (with a probability of 90%) of a given trend of |3| DU/yr starting at January 2008 (right panels: top, middle and bottom, respectively).



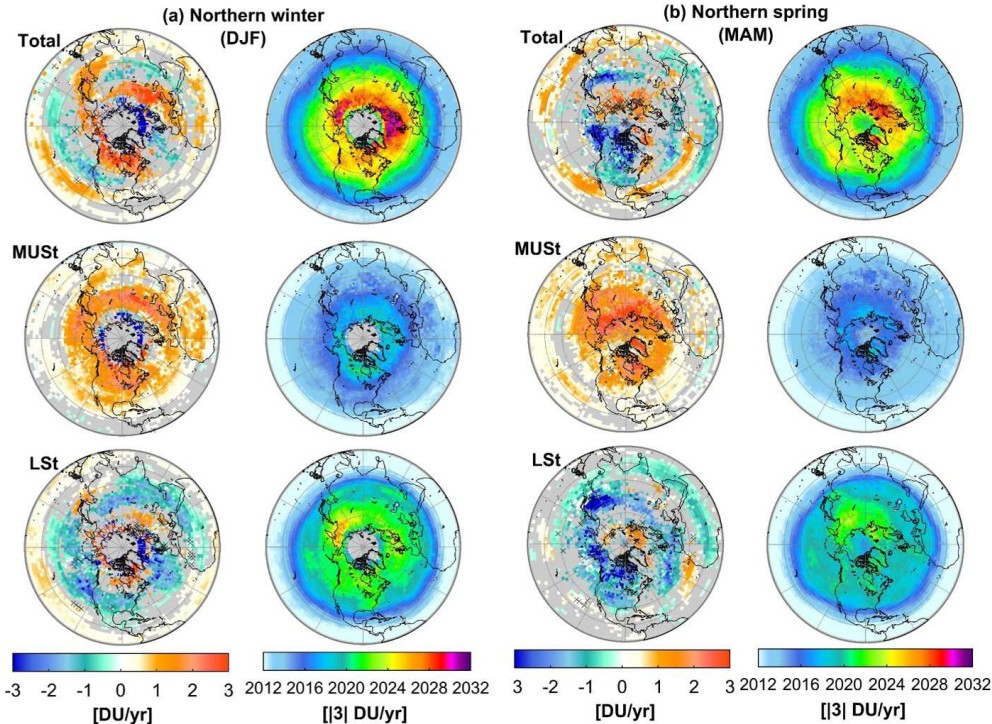


**Fig. 11:** Same as Fig. 10 but (a) for the winter (DJF) and (b) for the spring (MAM) of the northern Hemisphere.






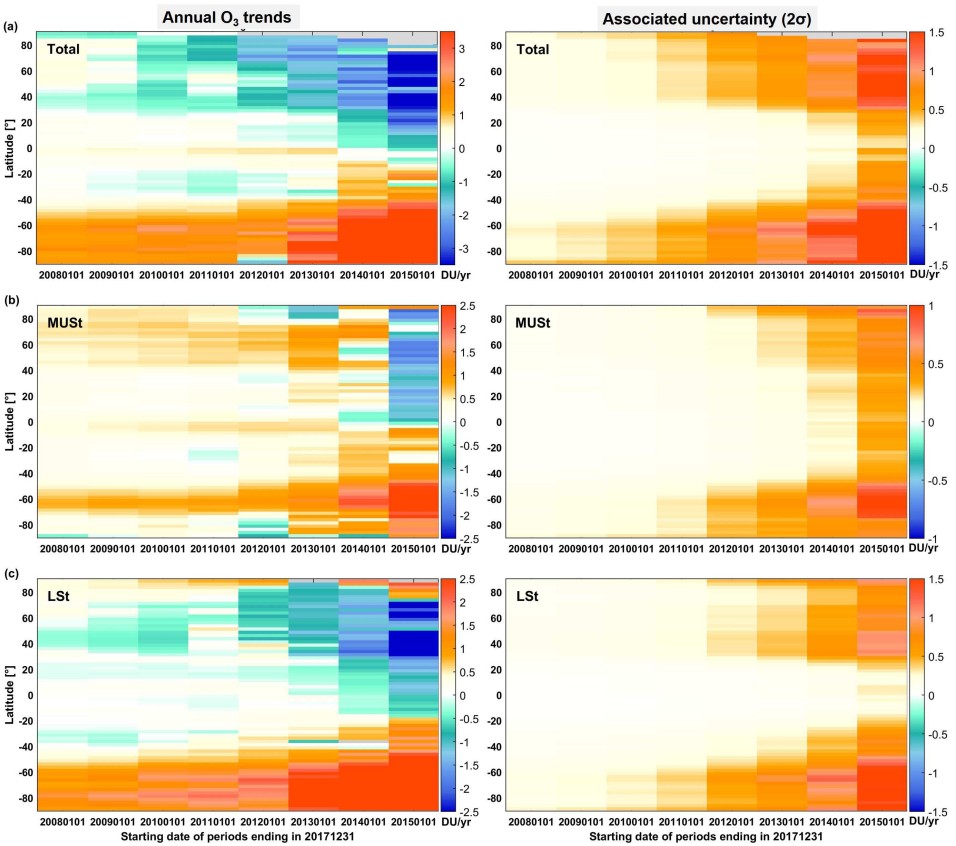


**Fig.12:** Evolution of estimated linear trend (DU/yr) and associated uncertainty (DU/yr; in the 95% confidence level) in (a) total, (b) MUSt and (c) LSt $O_3$ columns, as a function of the covered IASI measurement period ending in December 2017, with all natural contributions estimated from the whole IASI period (2008-2017). Note that the scales are different between the columns.






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
