# Peer review of "Is the recovery of stratospheric O3 speeding up in the Southern Hemisphere? An evaluation from the first IASI decadal record (2008-2017)"

_Atmospheric Chemistry and Physics, 2019_

## Referee Comment (RC1) · Anonymous Referee #2 · 21 May 2019

First review of manuscript entitled "Evidence from IASI of a speeding up in stratospheric O3 recovery in the Southern Hemisphere contrasting with the Decline in the Northern Hemisphere," by C. Wespes et al.

This manuscript describes ozone trends and variability in the IASI record from 2008 through 2017. A multiple linear regression model is used to isolate various natural sources of variability from the anthropogenic trend caused by CFCs, represented over this time period as a linear function. IASI measures ozone in four broad vertical ranges. Here the middle to upper stratospheric layer and the lower stratospheric layer are ex-

amined, as well as the total column ozone. This manuscript is largely an update of Wespes et al., 2016 but includes 4 more years of data.

The manuscript is well written, though there are occasions where the wording is confusing, likely due to language issues. The analysis is well structured and easy to follow, and the work is well referenced. The figures are clear and informative. However I suggest a major revision is needed because I question the results shown in Figure 12 on the ozone change rates, and the authors make some strong overall conclusions based on this analysis (including the title of the manuscript). If these issues can be addressed I believe the manuscript is a useful contribution to the ozone trends body of work and worthy of publication.

Major Comment: My primary comment concerns the analysis and conclusion that the ozone response to CFCs is changing in time. The authors base this conclusion on a series of linear fits over varying time periods, which show sharper trends (both positive and negative) in the most recent data relative to trends in the record from earlier start points. The series of trends is computed after the sources of natural variability, as fit over the full IASI time period to the most relevant proxies, are removed. Nevertheless there will still be variability in the time series that has not been perfectly captured by the regression model. If that variability has autocorrelation on a longer scale (months), a tendency for the data to be high or low at the beginning or end of the record, which might actually be due to uncaptured noise, will disproportionately affect the trend. If this is the case, such a variation at the end of the record will have successively more influence as the fit period gets shorter, as the end point of each fit is the same. If I understand correctly, the associated uncertainty plots in Fig. 12 tell us that each trend is different from zero trend at the 95% level, but that does not mean that the trend fit over the last 2 years is different from the trend fit over the last three years or last 4 years at the 95% level. For example in the SH high latitude LST the initial trend is ∼ 1 DU/yr with uncertainty of say 0.25 DU/yr (difficult to tell exact numbers from the contour plots) and the final trend is approaching 2.5 DU/yr with an uncertainty of close to 1.5

DU/yr, meaning the initial and final trends are not statistically significantly different or only barely so, depending on the exact numbers.

I believe a more appropriate approach would be to fit trend segments over the same length of time, with varying start and end points. The authors could compare the time evolution of trends over 2-yr segments, 3-yr segments, 4-yr segments and longer. The 2-yr segments would be the trend fit from 2008-2009, 2009-2010, 2010-2011, ..., 2015-2017. 3-yr segments would be 2008-2010, 2009-2011, ..., 2014-2017 and 4-yr 2008-2011, 2009-2012, ..., 2014-2017, and so on. In this way both the start and end point will vary, and each fit has the same length, such that the uncertainty is similar across the fits. If the results show consistent changes in time in the fit trends that are greater than the inherent uncertainty, this would indicate a change may be taking place. As the segments get longer (4-yr +) the change in trend will be less from segment to segment, but so will the uncertainty threshold that must be met to show significant change. So the authors can check for consistency in the trends within each segment length vs. time and consistency between 2-yr, 3-yr, 4-yr etc... segment results to determine if there is a shift in the ozone change rate.

I also believe showing some example time series of the data being fit, after the other variations have been removed, would be very useful in this particular analysis.

Finally, when doing this analysis, is the VPSC term also removed, or is this term considered part of the ozone response to CFCs and thus left in the time series? Similarly, in reference to the jump in the data in September 2010, although this may be small relative to the full trend, does this jump influence the results of the time dependent trend analysis shown in Fig. 12, or has it's effects been removed before fitting these trends?

Additional Comments:

Can the authors say more about the difference between fitting a daily record and a monthly mean record? I know this was addressed in the 2016 paper, but I am particularly interested in the error analysis. Is the daily autocorrelation similar to the monthly

autocorrelation? For long-term trends, the uncertainty is more impacted by correlations in the residual on longer time scales rather than day to day variations. Is the lag-1 autocorrelation term used to scale the uncertainty similar when considering daily data and monthly data?

Although I appreciate not wanting to add too much to the paper, I think it would help the reader to repeat the basic equations defining the multivariate model in this paper. At different times three different papers are referenced for equations concerning the model. I think it would be easier to just include all relevant equations in this paper, including the normalization equation.

Very little is said about the seasonal cycle, though the model description includes terms for the annual and 6-month harmonics (pg 5.). Can the authors comment on the seasonal cycle, and particularly do they see the seasonal cycle interacting with EPF and VPSC, which are both also correlated and look very seasonal in nature. Similarly on the interaction between EPF and VPSC, in Fig. 7a in the NH high latitudes the ozone variability explained by the proxies for EPF and VPSC are similar and well above the variability of the actual IASI ozone. Is this another way of showing that the two terms falsely depict variability that isn't in the actual data, but that variability cancels when the terms are added? Have the authors tried fitting to one or the other of the terms, rather than both terms? Particularly in the Austral Spring, where the authors believe the VPCS signal is real, is the amplitude of that signal sensitive to whether or not EPF and/or the seasonal cycle are fit?

Can the authors discuss comparisons between IASI total ozone and other sources of satellite total ozone measurements? It is difficult to compare trend values presented here with previous studies (Weber et al for example) because of the different time periods fit, and zonal mean vs high spatial resolution gridded trends. Have IASI total ozone trends been directly compared to trends from any of the other total ozone satellite records? It would be very useful to also see how the data themselves compare in total ozone, either through reference to previous work or in a comparison plot in this

manuscript.

Can the authors address how the seasonal averages are constructed? In particular, the authors specifically investigate the JJA trends over the South Pole and Antarctica, but it appears from Fig. 4a there is very little is any coverage in the deep winter at polar latitudes, but that coverage increases with latitude towards the equator. Are the JJA averages for each grid point made with any available data, or is a threshold set, and does the coverage vary with latitude in the polar regions in Figure 10 and 11?

Detailed Comments/Language/Typos

The use of the absolute value signs around the trend values was a bit confusing. I can see this when talking about the amount of time needed to detect a trend of $|x|$ DU yr-1 because this can be a positive or negative trend, but in other cases the authors state the trend is positive or negative, and in that case it is unclear why the absolute value designation is needed. For example on page 15, the absolute value bars are not needed in lines 561 and 564. In line 591, is this a positive trend of 1.5 DU/yr or do you mean positive or negative? If the authors do not mean to say this value can be positive or negative, I would suggest removing the absolute value bars and just stating positive or negative (such as in line 594, positive is stated so the bars can be removed, to me at least the bars imply positive or negative).

L12 should this be > 25hPa or < 25hPa? Since the units are in hPa I suggest it is < as in 25 hPa and lower pressures.

L34 in a lesser -> to a lesser L41 introduce O3 after ozone L43 gas. In the stratosphere . . . L45 for regulating -> to regulate L45 introduce chlorofluorocarbons here, at first use of CFCs L47-48 suggest These latter are the origin of the massive -> CFCs cause L46-54: In general, I don't think the timing is correct is this introduction to the phase out of the CFCs. At the time the Vienna Convention was ratified, and the MP for that matter, it was not yet proven that CFCs were the cause. The Vienna Convention was ratified based on the theory that CFCs could cause ozone destruction; I don't believe

the Farman paper was even released yet. All this to say, even though this is just an introductory paragraph I think it is important to be precise on the history, the implication in the wording is that the ozone hole was discovered first and everything else was a reaction to that discovery.

L56 Suggest removing first phrase, and start sentence as A recovery from . . .

L59 This is decline of CFCs in the stratosphere, correct? L61 confirmed -> identified L67 polar region -> polar regions L68 No reliable estimates of long-term trend -> Statistically significant long-term recovery in total O3 column on a global scale has not yet been observed, likely because . . . L71 low -> lower L75 I believe there are other references here as well. Check Wargan, K., C. et al. Recent decline in lower stratospheric ozone attributed to circulation changes. Geophys. Res. Lett., 45, no. 10, 5166-5176, doi:10.1029/2018GL077406.

L81 controversy -> uncertainty L82 sensitive -> difficult L109 applied on -> applied to L110 remove 'of' L172 and contrasts with -> rather than L178-180 the effect of the jump is found small enough to explain the trend? I'm not sure what the authors mean here.

L192 In order to unambiguously -> In an effort to unambiguously (we try to separate unambiguously, but it is never perfect)

L209 of the mixing L270-272 I'm not sure what the authors are trying to say here. Including the equations would help here. There is already a seasonal cycle in the original model, so it is not clear how the seasonal terms are added. Is this the equivalent of 4 separate runs, one for each season? Equations would also clarify how the seasonal MLR is used after the annual MLR is run. L285-288 suggest for clarity not switching the order of the reported results, in L288 LSt goes first and in 291 MUSt is reported first.

L302 counteracted -> counteracting (this may occur in other places as well in the text).

L 321 suggest adjusted signal of the proxies -> reconstructed proxies L333 shows up

as a typical . . . L347 MUSt, (remove 'n') L360 records -> values L392 deployment -> formation L414 remove 'have' L460 in the case of prolonged . . . L555 I do not see polar trends reaching 2.5 DU/yr in the MUSt? The trends are positive in the NH pole but negative over Antarctica, and the scale only goes to 2 DU/yr.

L560 The authors call out the similarity between the MUSt and LSt with both showing high positive trends at southern polar latitudes, but again at the pole the MUSt trend appears negative, though the trends at southern high latitudes are positive. This description seems a bit confusing and doesn't seem to match Figure 8.

L596 an additional $\sim$ 7 years L599 suggest The longer required measurement periods at high latitudes is due to the larger residuals in the regression fits (i.e. largest sigma e) at these latitudes (see Fig 4 a and b). L613 is there a reason the authors occasionally switch to DU per decade? If not, I suggest keeping DU per year. At first I could not understand why such a large value of 15 was used, then I saw it was DU per decade. L623-624 again it seems the increase in total ozone at high southern latitudes is dominated by the LSt result over the pole though both layers contribute in the latitude bands surrounding Antarctica, comparing to the results in Fig. 8.

L652 summer -> austral winter L674 over Antarctica (remove 'the') L686 what makes the negative trends here unrealistic?, It seems that the large positive trends off the coast of Antarctica have a similar detection length. I see that there is a bit more uncertainty in the fit in the negative trend region, but to say they are unrealistic requires more specific evidence, such as a time series showing the failure of the fit. I suggest the authors either provide more evidence or simply note that the area of higher negative trends is associated with a higher residual from the model. Could it also be something that is happening in the troposphere that is affecting the total ozone trend.

L696 Salomon -> Solomon

L705 This is just a suggestion, but to make the interpretation for the reader easier, could the authors provide the relevant IASI mean ozone values (or climatological values) so

the readers can translate between DU/yr and % per dec when comparing results from other studies.

L766 suggest However, a longer period of IASI measurements is needed to unequivocally demonstrate a positive trend in the IASI record.

L775 additional measurements for the trend to be unequivocal. L781 suggest These results verify the efficacy of the ban on ozone depleting substances imposed by the Montreal Protocol and it's amendments throughout the stratosphere . . ..

L788 and it likely -> which likely L807 in the near future L809 extent -> extend

---

## Referee Comment (RC2) · Anonymous Referee #1 · 11 Jun 2019

In their manuscript, Catherine Wespes and coworkers investigate stratospheric ozone changes from 10 years of global IASI satellite measurements. This is an important topic and the availability of IASI measurements over a sufficiently long period provides now an important new data set with high spatial and temporal coverage. The study is performed carefully with the required statistical rigor. The manuscript is well written and I recommend publication in Atmos. Chem. Phys. after consideration of the following comments (which in particular for the two general comments may require more than just minor corrections):

[Figure]

I have one general comment that relates to the simultaneous use of the linear trend with a VPSCxEESC proxy in the multi linear regression: How well can the linear O3 trend be determined at high latitudes (in winter/spring) when part of this change (through the EESC factor) is already included in the MLR? The combined effect of EESC and linear trends in polar regions is briefly addressed at L.690, but I could not find a discussion on the effect on the trends. This is in particular important in the light of the strong statements made: "To the best of our knowledge, these results represent the first detection of a significant recovery in the stratospheric and the total O3 columns over the Antarctic from one single satellite dataset."

I have to say that I am skeptical about the robustness of the speeding up of the trends in recent years, given that these trends are evaluated over really short periods only. Although the authors have done the analysis with statistical rigor, linear trends over periods as short as 2 years (2015-2017) are prone to changes in atmospheric dynamics and circulation (or other factors) that may not be perfectly captured by the MLR proxies. I (strongly) suggest that the authors consider a more careful wording in the conclusions and abstract, stating the evidence for the speeding up of the trends, but also the inherent uncertainties.

Specific comments:

L.72: Is this true for both hemispheres, or only NH?

L.83: "sensitive" does not seem the right word here. Sensitive to what?

Section 2.2: It would be good to have an explicit formula for the MLR included here, in addition to the reference to eq. (1) in Wespes et al. (2016).

L.210: A few more words on the GEO and PV proxies would be helpful. Although L.372 states that their contribution is generally small, their use in ozone trend studies is not common practice, so some reference to their purpose and how and why they improve the fit is justified. Are these proxies lat/lon dependent?

L.357++: SF: energetic particle precipitation (solar protons and also electrons) can also lead to enhanced ozone destruction in the MUSt through NOx catalysed cycles. The main effect of a solar proton event in the MUSt is actually to decrease O3 (and only to second order to decrease O3 destruction).

L.380++: EPF: I am surprised that the correlation of IASI O3 with EPF is small at low latitudes: Weber et al. (2011) note a rather strong anti-correlation between tropical total ozone and extra-tropical EPF.

L.474: suggestion "N.H. mode" -> NAO

L.480: Just as a note: It may also be that large O3 changes impact on the AAO

L.514: "if the influence of ENSO on stratospheric O3 measurements has been reported": the word "if" seems a bit out of place here as clearly the influence of ENSO on stratospheric O3 has been reported in the cited studies.

Technical corrections

L.233: "EFP" -> "EPF"

Additional Reference:

Weber et al., The Brewer-Dobson circulation and total ozone from seasonal to decadal time scales, Atmos. Chem. Phys., 11, 11221–11235, 2011.

---

## Author Comment (AC2) · 22 Jul 2019

The comment was uploaded in the form of a supplement:
https://www.atmos-chem-phys-discuss.net/acp-2019-206/acp-2019-206-AC2-supplement.pdf

---

## Author Response (AR1)

**Response to Referee #1:**

We are grateful to the referee for her/his careful reading of the manuscript and for her/his corrections and suggestions. Responses to each individual comment that has been quoted […] are given here below.

**General comments**

1/ *[I have one general comment that relates to the simultaneous use of the linear trend with a VPSCxEESC proxy in the multi linear regression: How well can the linear O3 trend be determined at high latitudes (in winter/spring) when part of this change (through the EESC factor) is already included in the MLR? The combined effect of EESC and linear trends in polar regions is briefly addressed at L.690, but I could not find a discussion on the effect on the trends. This is in particular important in the light of the strong statements made: "To the best of our knowledge, these results represent the first detection of a significant recovery in the stratospheric and the total O3 columns over the Antarctic from one single satellite dataset."]*

As already found in previous studies and stated in the manuscript, "the PSC volume is multiplied by the EESC to account for the changes in the amount of inorganic stratospheric chlorine that activates the polar ozone loss". In other words, the EESC factor is used to decrease the "efficiency" of the VPSC in activating the $O_3$ loss.

Actually, there is no possible confusion in the MLR between the linear trend and the VPSC x EESC proxy that is non-linear by nature given the strong oscillations in VPSC. The effect of the change in EESC on the amplitudes of the annual oscillations in VPSC which variate from year to year is very weak with, hence, no tendency detectable at all in the VPSC x EESC proxy (see Figure 1 here below). Therefore, it could not compensate the linear trend adjustment at all.

2/ *[I have to say that I am skeptical about the robustness of the speeding up of the trends in recent years, given that these trends are evaluated over really short periods only. Although the authors have done the analysis with statistical rigor, linear trends over periods as short as 2 years (2015-2017) are prone to changes in atmospheric dynamics and circulation (or other factors) that may not be perfectly captured by the MLR proxies. I (strongly) suggest that the authors consider a more careful wording in the conclusions and abstract, stating the evidence for the speeding up of the trends, but also the inherent uncertainties.]*

The speeding up has been investigated by removing the natural variability adjusted over the whole IASI period in order to avoid the effect of short trend-like segments in natural variations on the trend determination.
However, it is true that the uncaptured variability from the MLR performed over the full IASI period might disproportionately affect the estimated trends over varying time periods, but, so might be the calculation of the associated uncertainty, accordingly. This is specifically addressed in Fig.12 of the paper that illustrates the time evolution of both trends and associated uncertainties over varying time periods.
We agree, however, that the comparison of trends calculated over different lengths of time period is not straightforward because the statistical error is not comparable across the fits. This is addressed in Figure 2 here below that represents the minimum amplitude of the estimated trend, by subtracting the associated uncertainty (accounting for the autocorrelation in the noise residuals) from the linear trend; it still shows the significant increase in $O_3$ change rate across the fits.
Another approach, as suggested by Referee #2, would consist in considering successive time segments of same length. Nevertheless, here again, the uncaptured variability might induce different systematic errors between the successive segments, e.g. in case of "trend-like" noise over a specific segment. The choice of the segment length is also complicated by limitations (long segments would smooth the progressive acceleration, while short segment would induce larger uncertainty; the jump in September 2010 in the IASI dataset would misrepresent the trend calculated over short segments that encompass the jump period).

Finally, we believe that Fig.12 of the paper is the best alternative to represent the progressive acceleration in the $O_3$ recovery. Note also that we now consider the autocorrelation in the noise residuals in the uncertainty estimation illustrated in Fig.12.

Nevertheless, we agree that the IASI period is still relatively short to compare trends over successive segments of same length that are long enough to reduce the uncertainty.

Therefore, as suggested, we use, in the revised version, a more careful wording about the speeding up of the $O_3$ trends through the revised manuscript, especially in the abstract, in Section 4.4 and in the conclusions. For example, one can read now at the end of the abstract: "Additional years of IASI measurements would, however, be required to confirm the $O_3$ change rates observed in the stratospheric layers over the last years" and at the end of Section 4.4: "Nevertheless, we calculated that additional years of IASI measurements would help in confirming the changes in $O_3$ recovery and decline over the IASI period (e.g. ~ 4 additional years are required to verify the trends calculated over the 2015-2017 segment in the highest latitudes in LSt). In addition, a longer measurement period would be useful to derive trends over successive segments of same length that are long enough to reduce the uncertainty, in order to make the trend and its associated uncertainty more comparable across the fit."

The title of the manuscript has also been changed accordingly to: "Is the recovery of stratospheric $O_3$ speeding up in the Southern Hemisphere? An evaluation from the first IASI decadal record".

An alternative to that title would be: "First signs of a speeding up of stratospheric $O_3$ recovery in the Southern Hemisphere, contrasting with a decline in the Northern Hemisphere, as seen from IASI".

Finally, we have also found a bug in the calculation of the estimated trends through the manuscript. We apologize for this. The overall conclusions remain unchanged but the figures 8 to 12, and the numbers given in the text have been corrected accordingly.

**Specific comments**

*1/ [L.72: Is this true for both hemispheres, or only NH?]*

Ball et al. (2018) reports a decline in lower stratospheric $O_3$ between 60°S and 60°N. The polar regions are not included in that study due to limited latitude coverage of instruments merged in the data composites.

*2/ [L.83: "sensitive" does not seem the right word here. Sensitive to what?]*
Changed to "difficult".

*3/ [Section 2.2: It would be good to have an explicit formula for the MLR included here, in addition to the reference to eq. (1) in Wespes et al. (2016).]*

The MLR and the normalization equations are now included in the revised paper at the start of Section 2.2.

*4/ [L.210: A few more words on the GEO and PV proxies would be helpful. Although L.372 states that their contribution is generally small, their use in ozone trend studies is not common practice, so some reference to their purpose and how and why they improve the fit is justified. Are these proxies lat/lon dependent?]*

The use of the GEO and PV proxies is inherited from previous papers (e.g. Knibbe et al., 2014; Wespes et al., 2017) to account for the impact of tropopause height and of the mixing of tropospheric and stratospheric air masses, in particular, on the LSt $O_3$ variations. Their contributions into the LSt $O_3$ variations are found minor due to correlations with the annual harmonic term, as expected, but the proxies are kept in the MLR

for completeness. They are lat/lon dependent (2.5°x2.5° gridded; this is now mentioned in the revised Table 1), hence, their gridded adjusted coefficients are not comparable on a global basis; only the adjusted signals can be compared.

*5/ [L.357++: SF: energetic particle precipitation (solar protons and also electrons) can also lead to enhanced ozone destruction in the MUSt through NOx catalysed cycles. The main effect of a solar proton event in the MUSt is actually to decrease O3 (and only to second order to decrease O3 destruction).]*

Added as suggested. Note that the role of the solar proton event on the decrease of $O_3$ destruction, as mentioned in the paper, refers to the LSt where $NO_x$ decrease active chlorine and bromine.

*6/ [L.380++: EPF: I am surprised that the correlation of IASI O3 with EPF is small at low latitudes: Weber et al. (2011) note a rather strong anti-correlation between tropical total ozone and extra-tropical EPF.]*

Weber et al. found a negative correlation between tropical total ozone and extra-tropical EPF at lower latitudes throughout the winter and early spring, while it goes to zero by early summer. On an annual basis, Fig. 5 of the paper shows a weak but negative contribution (up to ~ -5 DU) onto $O_3$ variations. The negative sign which indicates an opposite response in $O_3$ to change in EPF is in agreement with the negative correlation, but the absolute value of the "regression" coefficient does not refer to the absolute value of the "correlation" coefficient; it indicates how much the proxy explains/contributes to the $O_3$ variations, while the absolute value of the correlation coefficient (as shown in Weber et al., 2011) indicates the degree of linearity between 2 variables.

The weak adjusted negative regression coefficients for EPF might result from correlation/compensation effect between the annual cycle and EPF. Despite the year-to-year variations in the EPF proxy, which limit the compensation effect with the 1-yr harmonic term, correlation between the two covariates is expected given the annual oscillations in EPF. This is illustrated in Figure 3 below that compares the global distribution of the fitted coefficient for the 1-yr harmonic term with or without EPF included in the MLR. The global distributions are quite similar with absolute differences (< 5 DU) lower than the EPF regression coefficient, indicating a good overall discrimination, except at the tropics where the EPF contribution is the lowest. Hence, the compensation effect between the 1-yr term (that is the main contributor to $O_3$ variations) and EPF might underrepresent its contribution at the Tropics. Note however that the correlation between the EPF and 1-yr terms is taken into account in their associated uncertainties.

Some words of caution have been added in the revised Section 3 about a likely compensation between the annual harmonic term and the EPF proxy that also shows an annual oscillation in nature:
"Furthermore, given the annual oscillations in EPF, compensation by the 1-yr harmonic term (eq. 1, Section 2) is found (data not shown), but it remains weaker than the EPF contribution (data not shown), in particular at high latitudes where the EPF contribution is the largest."

The Weber et al. (2011) reference has been added in the revised version.

*7/ [L.474: suggestion "N.H. mode" -> NAO]*

Changed as suggested.

*8/ [L480: Just as a note: It may also be that large O3 changes impact on the AAO]*

We apologise but we do not understand what the referee means here.

9/ *[L.514: "if the influence of ENSO on stratospheric O3 measurements has been reported": the word "if" seems a bit out of place here as clearly the influence of ENSO on stratospheric O3 has been reported in the cited studies.]*

Changed to: "Indeed, the influence of ENSO on stratospheric $O_3$ measurements has already been reported in earlier studies (…), but it is the first time that …"

**Technical corrections**

*[L.233: "EFP" -> "EPF"]*

Corrected

**Figures**

[Figure]

**Figure 1:** Normalized proxies as a function of time for the period covering January 2008 to December 2017 for the volume of polar stratospheric clouds multiplied or not by EESC and accumulated over time for the north and south hemispheres (VPSC-N and VPSC-S).

[Figure]

**Figure 2:** Evolution of estimated linear trend minus the associated uncertainty accounting for the autocorrelation in the noise residual (DU/yr; in the 95% confidence level) in (a) the total, (b) the MUSt and (c) the LSt O₃ columns (top to bottom panels, respectively), as a function of the covered IASI measurement period ending in December 2017, with all natural contributions estimated over the full IASI period (2008-2017).

[Figure]

**Figure 3:** Global distribution of the annual regression coefficient estimates for the 1-yr harmonic term ($\sqrt{a_1^2 + b_1^2}$, in DU) in LSt obtained from the annual MLR without or with EPF (left and right panels, respectively).

**Response to Referee #2:**

We are grateful to the referee for her/his very careful reading of the manuscript and for her/his constructive comments and suggestions. Responses to each individual comment that has been quoted […] are given here below.

**General comments**

*1/ [This manuscript is largely an update of Wespes et al., 2016 but includes 4 more years of data.]*

This manuscript is indeed built on previous IASI studies, but we hope that the referee will appreciate that it is actually more than an update of Wespes et al., 2016, insofar as the regression model is more complex and here adapted to stratospheric studies with the inclusion of specific proxies (accounting for the aerosols, the volume of PSC and the Eliassen-Palm flux), and as the analysis is now performed at the global scale, not on a zonal basis, which allows us to better demonstrate the added value of the IASI dataset.

*2/ [The manuscript is well written, though there are occasions where the wording is confusing, likely due to language issues]*

We are grateful to the referee for suggesting a series of English style corrections in her/his technical comments below. They have all been included in the revised paper.

**Major comments**

*1/ [My primary comment concerns the analysis and conclusion that the ozone response to CFCs is changing in time. The authors base this conclusion on a series of linear fits over varying time periods, which show sharper trends (both positive and negative) in the most recent data relative to trends in the record from earlier start points. The series of trends is computed after the sources of natural variability, as fit over the full IASI time period to the most relevant proxies, are removed. Nevertheless there will still be variability in the time series that has not been perfectly captured by the regression model. If that variability has autocorrelation on a longer scale (months), a tendency for the data to be high or low at the beginning or end of the record, which might actually be due to uncaptured noise, will disproportionately affect the trend. If this is the case, such a variation at the end of the record will have successively more influence as the fit period gets shorter, as the end point of each fit is the same.]*

The referee is right; the uncaptured variability might disproportionately affect the estimated trends when calculated over varying time periods, but, so might be the calculation of the associated uncertainty. This is specifically addressed in Fig.12 of the paper that illustrates the time evolution of both trends and associated uncertainties over varying time periods.
We agree that the comparison of trends calculated over different lengths of time is not straightforward and that considering successive time segments of same length would make the statistical error more comparable across the fits. Nevertheless, there are limitations in using successive identical segments as discussed below in response to the referee's suggestions. We note finally that the uncaptured variability might also induce different systematic errors between segments (of same or different lengths), e.g. in case of "trend-like" noise over a specific segment.
In order to address this issue, we now consider the autocorrelation in the noise residuals in the uncertainty estimation illustrated in Fig.12.

As discussed below in response to the two next referee's comments, we believe that the results shown in the revised Fig.12 of the paper are the best way to represent the time evolution of the trends over the 10- years IASI period. In addition to modifying the figure, we have taken care to better balance the findings through the revised manuscript, especially in the title, the abstract and the conclusions (see our responses here below).

Finally, we have also found a bug in the calculation of the estimated trends through the manuscript. We apologize for this. The overall conclusions remain unchanged but the Figures 8 to 12, and the numbers given in the text have been corrected accordingly.

*2/ [If I understand correctly, the associated uncertainty plots in Fig. 12 tell us that each trend is different from zero trend at the 95% level, but that does not mean that the trend fit over the last 2 years is different from the trend fit over the last three years or last 4 years at the 95% level. For example in the SH high latitude LST the initial trend is _ 1 DU/yr with uncertainty of say 0.25 DU/yr (difficult to tell exact numbers from the contour plots) and the final trend is approaching 2.5 DU/yr with an uncertainty of close to 1.5 DU/yr, meaning the initial and final trends are not statistically significantly different or only barely so, depending on the exact numbers.]*

We would like to point out that the exact numbers are given in Section 4.4 of the manuscript, specifically for the SH high latitude LSt: "In the LSt, a clear speeding up in the southern polar $O_3$ recovery is observed with amplitude ranging from ~1.5±0.4 DU/yr over 2008-2017 to ~5.5±2.5 DU/yr over 2015-2017 on latitudinal averages." Hence, the reader could appreciate that the initial and the final trends are statistically different from each other, despite the larger amplitude of the uncertainty over the shorter periods. This is further illustrated in Figure 1 here below which represents the lowest amplitude of the estimated trend, by subtracting, from the absolute value of the linear trend, the associated uncertainty that includes the autocorrelation in the noise residual.

The colorscale in the revised Fig.12 has been modified to avoid the saturation in order to address the comment on the lack of clarity. In addition, the uncertainty now accounts for the autocorrelation in the noise residuals and, hence, the uncertainty values are corrected accordingly throughout the manuscript.

*3/ [I believe a more appropriate approach would be to fit trend segments over the same length of time, with varying start and end points. The authors could compare the time evolution of trends over 2-yr segments, 3-yr segments, 4-yr segments and longer. The 2-yr segments would be the trend fit from 2008-2009, 2009-2010, 2010-2011,… 2015-2017. 3-yr segments would be 2008-2010, 2009-2011,…, 2014-2017 and 4-yr 2008-2011, 2009-2012, …, 2014-2017, and so on. In this way both the start and end point will vary, and each fit has the same length, such that the uncertainty is similar across the fits. If the results show consistent changes in time in the fit trends that are greater than the inherent uncertainty, this would indicate a change may be taking place. As the segments get longer (4-yr +) the change in trend will be less from segment to segment, but so will the uncertainty threshold that must be met to show significant change. So the authors can check for consistency in the trends within each segment length vs. time and consistency between 2-yr, 3-yr, 4-yr etc… segment results to determine if there is a shift in the ozone change rate.]*

We are grateful to the referee for this interesting suggestion. However, there are some limitations in using that approach:

- By fitting long segments, we would compare trends that are estimated over similar periods; i.e. for instance, 8-yr segments would imply comparing trends over 2008-2015 *vs* 2010-2017, which would smooth a progressive acceleration in the ozone change rate over the 10-year IASI period.
- By fitting short segments, we would induce a large uncertainty on the trend estimate (because of few data points and a hardly detectable trend from the noise) and, hence, less-conclusive results.
- The jump that occurs in September 2010 in the IASI dataset could over-represents disproportionally the estimated trends when they are calculated over short segments that encompass the jump period.

To follow the referee's suggestion, we have therefore investigated if the change rate in IASI $O_3$ could be inferred from segments that are long enough to lead low uncertainty and limit the jump effect. This is illustrated on Figure 2 here below that shows the trend evolution over 6-year, 7-year and 8-years segments in the LSt. Despite the smoothing of the trends over long periods, the progressive acceleration remains observed, especially in the Southern mid-latitudes. The results are also quite consistent with the revised Fig. 12, which gives more confidence in the speeding up observed in IASI LSt $O_3$.

Given the limitations discussed above, we believe that Fig.12 of the paper is the best way to represent the progressive acceleration in the $O_3$ recovery. Nevertheless, we agree that the IASI period is still relatively short to compare trends over successive segments of same length that are long enough to reduce the uncertainty. In addition, we calculate that the largest trend amplitudes derived over the last years of the IASI measurements would actually require a longer detection length than the covered time segments. Therefore, as suggested by the referee#1, we use, in the revised version, a more careful wording about the speeding up of the $O_3$ trends through the manuscript, especially in the abstract, in Section 4.4 and in the conclusions.

For example, one can read now at the end of the abstract: "Additional years of IASI measurements would, however, be required to confirm the $O_3$ change rates observed in the stratospheric layers over the last years" and at the end of Section 4.4: "Nevertheless, we calculated that additional years of IASI measurements would help in confirming the changes in $O_3$ recovery and decline over the IASI period (e.g. ~ 4 additional years are required to verify the trends calculated over the 2015-2017 segment in the highest latitudes in LSt). In addition, a longer measurement period would be useful to derive trends over successive segments of same length that are long enough to reduce the uncertainty, in order to make the trend and its associated uncertainty more comparable across the fit."

The title of the manuscript has also been changed accordingly to: "Is the recovery of stratospheric $O_3$ speeding up in the Southern Hemisphere? An evaluation from the first IASI decadal record".
An alternative to that title would be: "First signs of a speeding up in stratospheric $O_3$ recovery in the Southern Hemisphere, contrasting with a decline in the Northern Hemisphere, as seen from IASI".

4/ *[I also believe showing some example time series of the data being fit, after the other variations have been removed, would be very useful in this particular analysis.]*

We thank the referee for that suggestion. Some typical examples of gridded daily time series in the S.H. mid-latitudes in the LSt, after the fitted natural variations have been removed, are provided in the Figure 3 here below. The residuals clearly show positive trends. The fitted significant trends over varying periods ending in 2017 are superimposed. The trend values and associated uncertainties are also indicated for a conclusive evaluation of the significant $O_3$ change rate in stratosphere. While the speeding up is significant from the zonally averaged trends (see the revised Fig. 12 and the Figure 1 here below), it is more hardly but still detectable over individual gridded time series. Examples have been added in the revised Supplement.

5/ *[Finally, when doing this analysis, is the VPSC term also removed, or is this term considered part of the ozone response to CFCs and thus left in the time series? Similarly, in reference to the jump in the data in September 2010, although this may be small relative to the full trend, does this jump influence the results of the time dependent trend analysis shown in Fig. 12, or has it's effects been removed before fitting these trends?]*

All the adjusted proxies, including the VPSC term, have been kept fixed (or removed) in the trend analysis over varying time periods, so that any changes in the adjusted $O_3$ drivers (including in VPSCxEESC) over time do not influence the trend estimation. It is now clearly mentioned in the revised manuscript that VPSC is removed as well:
"…the ozone response to each natural driver *(including VPSC)* taken from their adjustment over the whole IASI period (2008-2017; Section 3, Fig.5) is kept fixed."

On the contrary, for consistency with Chapter 4, the jump found in the IASI data in September 2010 has not been removed from the trend analysis shown in Fig.12 and, hence, it could influence the trends calculated over the periods starting before the jump only (i.e. 2008-2017, 2009-2017 and 2010-2017). However, the jump is of positive sign and, hence, it does not contribute at all to the acceleration observed in the IASI $O_3$ change rates over the 10-year period. It would even mask it when comparing the trends estimated over periods starting before *vs* after the jump. This has been added in Section 4.4:
"The jump found in the IASI $O_3$ records on September 2010 (see Section 2.1) is not taken into account in the regression; hence, it might over-represent the trend estimated over periods that start before the jump only (i.e. 2008-2017, 2009-2017, 2010-2017)."

**Minor comments**

1/ *[Can the authors say more about the difference between fitting a daily record and a monthly mean record? I know this was addressed in the 2016 paper, but I am particularly interested in the error analysis. Is the daily autocorrelation similar to the monthly autocorrelation? For long-term trends, the uncertainty is more impacted by correlations in the residual on longer time scales rather than day to day variations. Is the lag-1 autocorrelation term used to scale the uncertainty similar when considering daily data and monthly data?]*

The autocorrelation coefficients at various lags corresponding to a daily mean record *vs* a monthly mean record were examined for the 2 stratospheric layers (cfr Figure 4 here below for the latitudinal distributions of the lag-1 to -4 autocorrelation terms in daily *vs* monthly data fitting in the MUSt). As expected, the lag-1 autocorrelation term appears to be the most important in all cases (daily and monthly) and is found to be much larger in the daily than the monthly mean records. This means that the correction of the uncertainty estimate, by the autocorrelations in the noise residual, is larger when adjusting daily data, i.e. the uncertainty associated to the fitted trend is much more impacted by the autocorrelation when fitting a daily record, but, as shown in the 2016 paper, it is compensated by a better quality adjustment, which, hence, reduces the amplitude of the uncertainty in daily *vs* monthly data records.

2/ *[Although I appreciate not wanting to add too much to the paper, I think it would help the reader to repeat the basic equations defining the multivariate model in this paper. At different times three different papers are referenced for equations concerning the model. I think it would be easier to just include all relevant equations in this paper, including the normalization equation.]*

The MLR and the normalization equations are now included in the revised paper at the start of Section 2.2.

3/ *[Very little is said about the seasonal cycle, though the model description includes terms for the annual and 6-month harmonics (pg 5.). Can the authors comment on the seasonal cycle, and particularly do they see the seasonal cycle interacting with EPF and VPSC, which are both also correlated and look very seasonal in nature. Similarly on the interaction between EPF and VPSC, in Fig. 7a in the NH high latitudes the ozone variability explained by the proxies for EPF and VPSC are similar and well above the variability of the actual IASI ozone. Is this another way of showing that the two terms falsely depict variability that isn't in the actual data, but that variability cancels when the terms are added? Have the authors tried fitting to one or the other of the terms, rather than both terms? Particularly in the Austral Spring, where the authors believe the VPCS signal is real, is the amplitude of that signal sensitive to whether or not EPF and/or the seasonal cycle are fit?]*

Correlation between the annual cycle and EPF is of course expected. In several previous papers, the harmonic terms are even used to adjust the effects of the Brewer-Dobson circulation in addition to the seasonal cycle of insolation, but then the interannual variability is not captured. However, the EPF and VPSC proxies show sufficiently year-to-year variations to limit the compensation effect between each other and with the 1-yr harmonic term.

In order to verify this, as suggested by the referee, an annual MLR without including EPF has been performed to better evaluate the possible discrimination between the EPF, VPSC and 1-yr harmonic terms. This is illustrated for LSt in Figure 5 here below that represents the global distributions of the adjusted coefficients for the 1-yr harmonic ( $\sqrt{a_1^2 + b_1^2}$ ) and the VPSC regression coefficients from the annual MLR without EPF *vs* the reference one. We show that the global distributions of the VPSC regression coefficients between the two MLRs are similar, which indicates a good discrimination between the two parameters on an annual basis. For the 1-yr coefficients, the overall global distributions look similar with, however, some expected but small differences relative to the EPF contribution, especially over the high latitudes where the EPF contribution is the largest. In addition, it is worth noting that the likely correlation between the VPSC, EPF and 1-yr terms is taken into account in their associated uncertainties.

Some words of caution have been added in the revised Section 3 about a possible correlation between the annual harmonic term and the EPF proxy:
"Furthermore, given the annual oscillations in EPF, compensation by the 1-yr harmonic term (eq. 1, Section 2) is found (data not shown), but it remains weaker than the EPF contribution, in particular at high latitudes where the EPF contribution is the largest."

We would like to point out that the likely correlation between VPSC and EPF was already mentioned in the paper in Sections 2.2 and 3 which describe the proxies and their adjustment: "Correlations between VPSC and EPF are possible since the same method is used to build these cumulative proxies". They can indeed compensate each other by construction given the opposite sign of their regression coefficients. However, we highlight the physical meaning behind the sign of their regression coefficients and the differences between the spatial distributions of their regression coefficients (see Fig.5 of the manuscript), which indicate a discrimination between these two variables.

On a seasonal basis, the austral spring is the period when VPSC is the largest and dominates over EPF in the S.H.; this is consistent with the role of PSCs on the polar $O_3$ depletion chemistry and the smallest EP influence due the formation of the $O_3$ hole, in comparison with the N.H. However, a compensation effect might indeed explain the large similar VPSC and EPF variability in the N.H. high latitudes in fall, as it was already mentioned in the paper: "The strong VPSC influence found at high northern latitudes in fall (Fig. 7a) are likely due to compensation effects with EPF as pointed out above."

The good discrimination in austral spring and the compensation effect in the N.H. fall are verified in the Figure 6 here below that compares the latitudinal distribution of the $2\sigma$ $O_3$ variability in VPSC, from the seasonal MLR with or without including EPF. The amplitude of the variation explained by VPSC are similar between the two seasonal MLRs in the Austral spring, while, not in the N.H. fall. The results in Figures 5 and 6 here demonstrate a good discrimination between the two covariates yearly and in the Austral spring.

In the revised version, we now mention:
"The strong VPSC influence found at high northern latitudes in fall (Fig. 7a) are due to compensation effects with EPF as pointed out above and verified from sensitivity tests (data not shown)."

Finally, we believe that it does not make sense to remove both the 1-yr harmonic term and EPF from the MLR model; the annual cycle that is caused by solar insolation which is the main driver of the observed $O_3$ variability will no longer be represented, which will lead to erroneous results.

4/ [*Can the authors discuss comparisons between IASI total ozone and other sources of satellite total ozone measurements? It is difficult to compare trend values presented here with previous studies (Weber et al for example) because of the different time periods fit, and zonal mean vs high spatial resolution gridded trends. Have IASI total ozone trends been directly compared to trends from any of the other total ozone satellite records? It would be very useful to also see how the data themselves compare in total ozone, either through reference to previous work or in a comparison plot in this manuscript.*]

Performing comparisons between $O_3$ trends derived from IASI *vs* other satellite instruments would be of course interesting for evaluating the inferred trends and the relevance of the current datasets to carry out trend studies. However, it is a significant endeavour that is beyond the scope of the present study. Actually, this will be specifically addressed in the frame of the recently started Ozone_CCI+ program where the IASI $O_3$ trends will be compared to those estimated from GOME-2 (both onboard the Metop platforms) over exactly the same time period and using the same MLR model/method. In that way, the bias resulting from different time periods, spatial/temporal samplings and trend calculations will be excluded.

5/ [*Can the authors address how the seasonal averages are constructed? In particular, the authors specifically investigate the JJA trends over the South Pole and Antarctica, but it appears from Fig. 4a there is very little is any coverage in the deep winter at polar latitudes, but that coverage increases with latitude towards the equator. Are the JJA averages for each grid point made with any available data, or is a threshold set, and does the coverage vary with latitude in the polar regions in Figure 10 and 11?*]

The distributions of seasonal trends provided in Fig. 10a and 11a of the paper do not correspond to averages; instead they represent the adjusted seasonal trend parameters for each grid cell (see our response to the technical comment [*L270-272*] below). It is true, however, that the coverage vary with latitude in the polar regions since only the daytime measurements are used in the paper (as mentioned in Section 2.1). This explains the gap (grey cells) over the polar regions during both austral and northern winters in Fig. 10a and 11a of the paper, in comparison with the other periods (Fig. 10b and 11b) and the annual trend distributions (Fig. 8 of the paper).

**Technical comments**

1/ [*The use of the absolute value signs around the trend values was a bit confusing. I can see this when talking about the amount of time needed to detect a trend of |x| DU yr-1 because this can be a positive or negative trend, but in other cases the authors state the trend is positive or negative, and in that case it is unclear why the absolute value designation is needed. For example on page 15, the absolute value bars are not needed in lines 561 and 564. In line 591, is this a positive trend of 1.5 DU/yr or do you mean positive or negative? If the authors do not mean to say this value can be positive or negative, I would suggest removing the absolute value bars and just stating positive or negative (such as in line 594, positive is stated so the bars can be removed, to me at least the bars imply positive or negative).*]

The consistency in using the absolute value bars has been checked through the manuscript. The absolute bars are now only used when discussing the detectability of a specified trend (i.e. when the trend can be of both positive and negative values); in other cases, the sign is specified.

2/ [*L12 should this be > 25hPa or < 25hPa? Since the units are in hPa I suggest it is < as in 25 hPa and lower pressures. L34 in a lesser -> to a lesser. L41 introduce O3 after ozone. L43 gas. In the stratosphere*

*L45 for regulating -> to regulate. L45 introduce chlorofluorocarbons here, at first use of CFCs. L47-48 suggest These latter are the origin of the massive -> CFCs cause. L46-54: In general, I don't think the timing is correct is this introduction to the phase out of the CFCs. At the time the Vienna Convention was ratified, and the MP for that matter, it was not yet proven that CFCs were the cause. The Vienna Convention was ratified based on the theory that CFCs could cause ozone destruction; I don't believe the Farman paper was even released yet. All this to say, even though this is just an introductory paragraph I think it is important to be precise on the history, the implication in the wording is that the ozone hole was discovered first and everything else was a reaction to that discovery. L56 Suggest removing first phrase, and start sentence as A recovery from... L59 This is decline of CFCs in the stratosphere, correct? L61 confirmed -> identified. L67 polar region -> polar regions. L68 No reliable estimates of long-term trend -> Statistically significant long-term recovery in total O3 column on a global scale has not yet been observed, likely because... L71 low -> lower. L75 I believe there are other references here as well. Check Wargan, K., C. et al. Recent decline in lower stratospheric ozone attributed to circulation changes. Geophys. Res. Lett., 45, no. 10, 5166-5176, doi:10.1029/2018GL077406. L81 controversy -> uncertainty. L82 sensitive -> difficult. L109 applied on -> applied to. L110 remove 'of'. L172 and contrasts with -> rather than*]

Thank you for these corrections. The text has been revised as suggested. Note in particular the following points:
- $O_3$ was already introduced in the abstract.
- The timing in the introduction has been corrected in the revised version. The Farman et al. paper was accepted (28 March 1985) just after the Vienna Convention (22 March 1985).
- Wargan et al. (2018) has been added in the introduction.

*3/ [L178-180 the effect of the jump is found small enough to explain the trend? I'm not sure what the authors mean here.]*

Changed to:
"The estimated amplitude of the jump is found to be relatively small in comparison to that of the decadal trends derived in Section 4, hence, it cannot explain the tendency in the IASI dataset. Therefore, the jump is not taken into account in the MLR."

*4/ [L192 In order to unambiguously -> In an effort to unambiguously (we try to separate unambiguously, but it is never perfect). L209 of the mixing*]

Done as suggested.

*5/ [L270-272 I'm not sure what the authors are trying to say here. Including the equations would help here. There is already a seasonal cycle in the original model, so it is not clear how the seasonal terms are added. Is this the equivalent of 4 separate runs, one for each season? Equations would also clarify how the seasonal MLR is used after the annual MLR is run.]*

As it is stated in the paper, the seasonal MLR replace the annual functions with 4 seasonal functions, i.e. by adjusting 4 coefficients (one for each seasonal functions for the main proxies, instead of only one coefficient per annual function in the annual MLR). Hence, in the seasonal MLR, the explanatory variables are split into four seasonal functions ($x_{spr}X_{norm,spr} + x_{sum}X_{norm,sum} + x_{fall}X_{norm,fall} + x_{wint}X_{norm,wint}$) that are simultaneously and independently adjusted. There is only one run (as for the annual MLR) with 4 adjusted parameters per proxy. Note that this is not to be confused with the seasonal cycle (harmonic terms) which is treated exactly the same way in both the annual and seasonal formulation of the MLR model (only one annual coefficient is adjusted for each harmonic function). Hence, the seasonal MLR is not equivalent to 4 separate runs. The seasonal MLR takes into account the different influence of the geophysical processes onto $O_3$ across the seasons, while the annual model is more constrained by the adjustment of year-round proxies which, hence, induces larger systematic errors.

The sentence has been rewritten in the revised version to:
"In the seasonal formulation of the MLR model, the main proxies ( $x_j X_{norm,j}$ ; with $x_j$ , the regression coefficient and $X_{norm,j}$ the normalized proxy) are split into four seasonal functions ( $x_{spr} X_{norm,spr} + x_{sum} X_{norm,sum} + x_{fall} X_{norm,fall} + x_{wint} X_{norm,wint}$ ) that are independently and simultaneously adjusted for each grid cell."

6/ [*L285-288 suggest for clarity not switching the order of the reported results, in L288 LSt goes first and in 291 MUSt is reported first. L302 counteracted -> counteracting (this may occur in other places as well in the text). L 321 suggest adjusted signal of the proxies -> reconstructed proxies. L333 shows up as a typical... L347 MUSt, (remove 'n'). L360 records -> values. L392 deployment ->formation. L414 remove 'have'. L460 in the case of prolonged...*]

Done as suggested.

7/ [*L555 I do not see polar trends reaching 2.5 DU/yr in the MUSt? The trends are positive in the NH pole but negative over Antarctica, and the scale only goes to 2 DU/yr. L560 The authors call out the similarity between the MUSt and LSt with both showing high positive trends at southern polar latitudes, but again at the pole the MUSt trend appears negative, though the trends at southern high latitudes are positive. This description seems a bit confusing and doesn't seem to match Figure 8.*]

Some cells were indeed characterized by trends of 2.5 DU/yr even if the color scale is saturated at 2 DU/yr for clarity. From Fig.8a of the manuscript, one can see that the trends in MUSt are positive almost everywhere, except over Antarctica, with the largest values over the northern polar region and around Antarctica for the S.H.
"(except over Antarctica)" has been added in the revised text to exclude this from the discussion.
Note that the Fig. 8 and the corresponding values given in the text have been revised to correct a bug, as mentioned above.

8/ [*L596 an additional _ 7 years. L599 suggest The longer required measurement periods at high latitudes is due to the larger residuals in the regression fits (i.e. largest sigma e) at these latitudes (see Fig 4 a and b). L613 is there a reason the authors occasionally switch to DU per decade? If not, I suggest keeping DU per year. At first I could not understand why such a large value of 15 was used, then I saw it was DU per decade. L623-624 again it seems the increase in total ozone at high southern latitudes is dominated by the LSt result over the pole though both layers contribute in the latitude bands surrounding Antarctica, comparing to the results in Fig. 8.*]

Done as suggested.

9/ [*L652 summer -> austral winter. L674 over Antarctica (remove 'the'). L696 Salomon -> Solomon*]

Done as suggested.

10/ [*L686 what makes the negative trends here unrealistic?, It seems that the large positive trends off the coast of Antarctica have a similar detection length. I see that there is a bit more uncertainty in the fit in the negative trend region, but to say they are unrealistic requires more specific evidence, such as a time series showing the failure of the fit. I suggest the authors either provide more evidence or simply note that the*

*area of higher negative trends is associated with a higher residual from the model. Could it also be something that is happening in the troposphere that is affecting the total ozone trend.*]

"unrealistic" has been replaced by "higher"; The large positive trends around Antarctica have a shorter detection length.

11/ [*L705 This is just a suggestion, but to make the interpretation for the reader easier, could the authors provide the relevant IASI mean ozone values (or climatological values) so the readers can translate between DU/yr and % per dec when comparing results from other studies.*]

The trend in IASI TOC is now given in %/dec as well.

12/ [*L766 suggest However, a longer period of IASI measurements is needed to unequivocally demonstrate a positive trend in the IASI record. L775 additional measurements for the trend to be unequivocal. L781 suggest These results verify the efficacy of the ban on ozone depleting substances imposed by the Montreal Protocol and it's amendments throughout the stratosphere... L788 and it likely -> which likely. L807 in the near future. L809 extent -> extend*]

Done as suggested.

**Figures**

[Figure]

**Figure 1:** Evolution of estimated linear trend minus the associated uncertainty accounting for the autocorrelation in the noise residual (DU/yr; in the 95% confidence level) in (a) the total, (b) the MUSt and (c) the LSt $O_3$ columns, as a function of the covered IASI measurement period ending in December 2017, with all natural contributions estimated over the full IASI period (2008-2017).

[Figure]

**Figure 2:** Evolution of estimated linear trend (DU/yr) in the LSt $O_3$ columns, over (a) 6-year, (b) 7-year and (c) 8 years segments of IASI measurements, with all natural contributions estimated over the full IASI period (2008-2017).

[Figure]

**Figure 3:** Example of gridded daily time series of $O_3$ measured by IASI in the LSt over the period 2008-2017 with all the contributions to $O_3$ variations adjusted from MLR over the full IASI period removed, except for the trend (in DU). The significant fitted trends calculated over varying time periods from a single linear regression are superimposed. The trend values with associated uncertainties (in the 95% confidence level; in DU/yr) are indicated.

[Figure]

**Figure 4:** Latitudinal distribution of the lag-1 to -4 autocorrelation terms in the noise residuals when fitting a daily mean (left panel) *vs* a monthly mean (right panel) record in the MUSt over 2008-2017.

[Figure]

**Figure 5:** Global distributions of the annual regression coefficient estimates (in DU) for the 1-yr harmonic term ($\sqrt{a_1^2 + b_1^2}$, top panels) and for the VPSC proxy (bottom panels) in LSt obtained from the annual MLR without or with EPF (left and right panels).

[Figure]

**Figure 6:** Same as Fig.7 of the paper for the austral spring periods (SON) in LSt, with, superimposed, the 2σ O₃ variability due to variations in VPSC from the seasonal MLR without EPF (dark green).

**List of relevant changes made in the manuscript:**

**Title:**

"Is the recovery of stratospheric $O_3$ speeding up in the Southern Hemisphere? An evaluation from the first IASI decadal record"

**Abstract:**

- **L. 14:** "(MUSt; <25hPa)"

- **L. 40-41:** "Additional years of IASI measurements would, however, be required to confirm the $O_3$ change rates observed in the stratospheric layers over the last years."

**Introduction:**

- **L. 48-56:** "In the 1980s, the scientific community motivated decision-makers to regulate the use of chlorofluorocarbons (CFCs), after the unexpected discovery of the springtime Antarctic ozone hole (Chubachi, 1984; Farman et al., 1985) that was suspected to be induced by continued use of CFCs (Molina and Rowland, 1974; Crutzen, 1974); The $O_3$ depletion was later verified from measurements at other Antarctic sites (e.g. Farmer et al., 1987) and from satellite observations (Stolarski et al., 1986), and explained by the role of CFC's on the massive destruction of $O_3$ following heterogeneous reactions on the surface of polar stratospheric clouds (Solomon, 1986; 1999 and references therein)."

**Section 2.1:**

- **L. 185-187:** "The estimated amplitude of the jump is found to be relatively small in comparison to that of the decadal trends derived in Section 4, hence, it cannot explain the tendency in the IASI dataset. Therefore, the jump is not taken into account in the MLR."

**Section 2.2:**

- **L. 200-213:** "we have applied to the 2.5°x2.5° gridded daily MUSt and LSt $O_3$ time series, a MLR model that is similar to that previously developed for tropospheric $O_3$ studies from IASI (see Wespes et al., 2017; 2018) but here adapted to fit the stratospheric variations:

$$O_3(t) = Cst + x_{j=1} \cdot trend + \sum\nolimits_{n=1:2}\left[a_n \cdot \cos(n\omega t) + b_n \cdot \sin(n\omega t)\right] + \sum_{j=2}^{m} x_j X_{norm,j}(t) + \varepsilon(t) \quad (1)$$

where $t$ is the number of days, $x_1$ is the trend coefficient in the data, $\omega$ = 2π/365.25, $a_n, b_n, x_j$ are the regression coefficients of the seasonal and non-seasonal variables and $\varepsilon(t)$ is the residual variation (assumed to be autoregressive with time lag of 1 day). $X_{norm,j}$ are the $m$ chosen explanatory variables, commonly called "proxies", which are normalized over the study period (2008 – 2017) with:

$$X_{norm}(t) = 2\left[X(t) - X_{median}\right]/\left[X_{max} - X_{min}\right] \quad (2)"$$

- **L. 286-289:** "In the seasonal formulation of the MLR model, the main proxies ( $x_j X_{norm,j}$ ; with $x_j$ the regression coefficient and $X_{norm,j}$ the normalized proxy) are split into four seasonal functions ( $x_{spr} X_{norm,spr} + x_{sum} X_{norm,sum} + x_{fall} X_{norm,fall} + x_{wint} X_{norm,wint}$ ) that are independently and simultaneously adjusted for each grid cell (Wespes et al., 2017)."

**Section 3:**
- **L. 388-389:** "… while it enhances the $O_3$ destruction in the MUSt through $NO_x$ catalysed cycles,…"

- **L. 419-422:** "Furthermore, given the annual oscillations in EPF, compensation by the 1-yr harmonic term (eq. 1, Section 2) is found, but it remains weaker than the EPF contribution (data not shown), in particular at high latitudes where the EPF contribution is the largest."

- **L. 429-432:** "The strong VPSC influence found at high northern latitudes in fall (Fig. 7a) are due to compensation effects with EPF as pointed out above and verified from sensitivity tests (not shown)."

**Section 4.1:**
- **L. 576-577:** "…the MUSt shows significant positive trends larger than 1 DU/yr poleward of ~35°N/S (except over Antarctica)."

- **L. 625-627:** "The longer required measurement period at high latitudes is due to the larger noise residuals in the regression fits (i.e. largest $\sigma_\varepsilon$ ) at these latitudes (see Fig.4 a and b)."

**Section 4.2:**
- **L. 650-652:** "In addition, the increase in total $O_3$ at high southern latitudes is dominated by the LSt, although both layers positively contribute around Antarctica, comparing to the trend distributions in Fig. 8."

**Section 4.3:**
- **L. 677-680:** "Here we investigate the respective contributions of the LSt and the MUSt to the TOC recovery over the Southern latitudes during spring and also during winter when the minima in $O_3$ levels occur in the MUSt (down to ~60 DU in polar regions), in comparison with the Northern latitudes."

**Section 4.4:**
- **L. 743-747:** "The linear trend term only is adjusted over variable measurement periods that all end in December 2017, by using a single linear iteratively reweighted least squares regression applied on gridded daily IASI time series after all the sources of natural variability fitted over the full IASI period are removed (typical examples of linear trend adjustment can be found in the Fig. S2 of the supplementary materials)."

- **L. 747-750:** "The discontinuity found in the MUSt IASI $O_3$ records on September 2010 (see Section 2.1) is not taken into account in the regression; hence, it might over-represent the trends estimated over periods that start before the jump (i.e. 2008-2017, 2009-2017, 2010-2017)."

- **L. 750-753:** "The zonally averaged results are displayed in Fig. 12 for the statistically significant total, MUSt and LSt $O_3$ trends and their associated uncertainty (accounting for the autocorrelation in the noise residuals; in the 95% confidence level) estimated from an annual regression."

- **L. 769-773:** "Overall, the larger annual significant trend amplitudes derived over the last few years of total, MUSt and LSt $O_3$ measurements, compared with those derived from the whole studied period (Sections 4.1 and 4.2) and from earlier studies, translate to trends that remain detectable over the increasing uncertainty associated to the shorter and shorter time segments (see Fig. S3 of the supplementary materials), especially in both LSt and total $O_3$ in the S.H."

- **L. 775-781:** "Nevertheless, we calculated that additional years of IASI measurements would help in confirming the changes in $O_3$ recovery and decline over the IASI period (e.g. ~ 4 additional years are required to verify the trends calculated over the 2015-2017 segment in the highest latitudes in LSt). In addition, a longer measurement period would be useful to derive trends over successive segments of same length that are long enough to reduce the uncertainty, in order to make the trend and its associated uncertainty more comparable across the fit.**"**

**Conclusion:**
- **L. 837-839:** "Even if the acceleration cannot be categorically confirmed yet, it is of particular urgency to understand its causes for apprehending its possible impact on the $O_3$ layer and on future climate changes."

- **L. 842-843**: "… which translate to trend values that would be categorically detectable in the next few years on an annual basis."

**Table 1:**
- **EPF:** "Vertical component of Eliassen-Palm flux crossing 100 hPa, averaged over 45°-75° for each hemisphere and accumulated over the 3 or 12 last months depending on the time period and the latitude (see text for more details) (*daily*)"
- **PV and GEO:** "… (2.5°x2.5° gridded) …"

**Throughout the manuscript:**
- The absolute bars are now only used when discussing the detectability of a specified trend (i.e. when the trend can be of both positive and negative values); in other cases, the sign is specified.

- Figures 8 to 12 have been corrected. Hence, the values referring to these figures have been changed accordingly in the revised text.

**Supplementary Materials:**

- Figures S2 and S3 with their associated description have been added in the Supplement.

[revised manuscript text omitted]

Figure S2 represents three typical examples of gridded daily time series of O₃ measured by IASI in the LSt over the period 2008-2017, after the natural variations fitted from MLR over the full

IASI period have been removed. The significant fitted trends calculated over varying time periods from a single linear regression are superimposed. The trend values with associated uncertainties (in the 95% confidence level; in DU/yr) are indicated.

**3 Figure S3**

Figure S3 illustrates the time evolution of the lowest amplitude of the estimated trends in (a) total, (b) MUSt and (c) LSt $O_3$ columns over varying time periods that all end in December 2017, by subtracting the associated uncertainty (accounting for the autocorrelation in the noise residual; in the 95% confidence level) from the absolute value of the linear trends.

 **Figure caption**

**Fig.S1:** Latitudinal distribution of (a) MUSt and (b) LSt $O_3$ columns as a function of time
observed from IASI (in DU; top panels), simulated by BASCOE (in DU; middle panels) and of
the IASI-BASCOE differences (in %; bottom panels). The black narrow in the difference panel
for the MUSt highlights a jump on 15[th] September 2010.

[Figure]

**Fig. S2:** Examples of gridded daily time series of $O_3$ measured by IASI in the LSt over the period 2008-
with all contributions to $O_3$ variations adjusted from MLR over the full IASI period removed, except
for the trend (in DU). The significant fitted trends calculated over varying time periods from a single
linear regression are superimposed. The trend values with associated uncertainties (in the 95% confidence
level; in DU/yr) are indicated.

[Figure]

**Fig. S3:** Evolution of estimated linear trend minus the associated uncertainty accounting for the autocorrelation in the noise residual (DU/yr; in the 95% confidence level) in (a) the total, (b) the MUSt and (c) the LSt O$_3$ columns, as a function of the covered IASI measurement period ending in December 2017, with all natural contributions estimated over the full IASI period (2008-2017).

---

## Editor Decision (ED1)

**Review**

**Major:**

Line 753-765: Estimating trends for just TWO years is not a good ozone recovery analysis. This would not yield any good or meaningful results in the recovery context. Both LT and MLR are very sensitive to number of years considered for the trend analysis. I have also done some sensitivity tests on the impact of number of years on the trend detection. Therefore, I would suggest you to remove the two-year trend analyses from this work to make your other analysis (10-year IASI data) more robust and appealing. I would like to remind you that one of the referees also had the same opinion/comment on this. Thank you.

**Minor:**

Line 1-2: please indicate the time period in the title, "the first IASI decade (2008-2018) record"

Line 32: What is meant by "decline is not categorical in to total o3"?

Line 32: "freezing the regression coefficient". What is this freezing?

Line 36: "trends over the year"? Usually it's over a time period. Not for a year!

Line 37: over the last years? How many years?

Line 39-40: it sounds that only IASI measurements are required to confirm the o3 change.

Line 51: the full-stop after the bracket

Line 62: but to be further delayed

Line 83: statistical approaches also use measurements or model results

Line 97-101: The sentence is too hard to understand. Please split and rephrase it

Line 103: what is high density here? Also join this small paragraph with the previous one.

Line 110: Usually we use the acronym LS for the lower stratosphere, not LSt

Line 119: "two years" is too short to estimate trends.

Line 121: speeding up of o3 changes? Or speeding up of ozone recovery?

Line 136: measurements of atmospheric composition? Or list some trace gases here.

Line 137: "2012, respectively"

Line 143: delete "For this study"

Line 150: what is the vertical resolution of o3 profiles?

Line 158: Please state the accuracy of the retrievals

Line 166: if 20% is the bias or accuracy, then how could that affect your statistical analyses presented, in terms of the ozone recovery estimation?

Line 177: "resulted from"

Line 181: full IASI period. Please state that particular period

Line 186: is it the "trends" or any other "tendency"

Line 192: flags determined or selected

Line 231: e.g. Wespes et al.

Line 303: it is a large range 25 to 95%. Could you please state where it reproduces the best and worst?

Line 322: delete "and discussed in this paper"

Line 336: "b, respectively,"

Line 340: "The ozone variability (i.e. 2-sigma)"

Line 368: Influence of QBO30 is greater, due to its altitude dependence?

Line 397: Yes, this has also been discussed in

*Roscoe, H. K. and Haigh, J. D.: Influences of ozone depletion, thesolar cycle and the QBO on the Southern Annular Mode, Q. J.Roy. Met. Soc., 133, 1855–1864, 2007.* and in Kuttippurath et al., 2013 (you have already cited this publication in this article).

Line 437-442: Please split this sentence, too long to comprehend

Line 458: ", which"

Line 489: "Asia that matches…"

Line 560: I do not understand this 80 DU here.

Line 620: 0.90; is it 90% CI level

Line 661, 663: hyphen

Line 674: A correction: Kuttippurath and Nair (2017) discussed the vertical trends in ozone, not only total column ozone. In addition, they also discussed the ozone trends in summer months. A recent study from the same authors reported recovery signatures even at the saturation altitudes (doi:10.1038/s41612-018-0052-6).

Line 805: You were discussing about 90% in previous pages. Now its 95%?

Line 808: "to unequivocally"

Line 808: Only 2- years are required? What is the basis for this statement?

---

## Author Response (AR2)

**Response to Editor:**

We thank the Editor for his reading of the manuscript and his suggestions. Our responses are given here below.

**Major comment:**

Line 753-765: Estimating trends for just TWO years is not a good ozone recovery analysis. This would not yield any good or meaningful results in the recovery context. Both LT and MLR are very sensitive to number of years considered for the trend analysis. I have also done some sensitivity tests on the impact of number of years on the trend detection. Therefore, I would suggest you to remove the two-year trend analyses from this work to make your other analysis (10-year IASI data) more robust and appealing. I would like to remind you that one of the referees also had the same opinion/comment on this. Thank you.

It is true that estimating O3 trends on short timescales could lead to non-significant results. However, we would like to recall that all the trend analysis done here rely on the full 10-year dataset of IASI (2008-2017) and that only the speeding up is based on shorter periods (the shortest being 3-years; over 2015-2017). Our analysis has been done carefully by removing the natural variability that has been adjusted on the 10-years of IASI; this is important in particular to avoid the effect of short trend-like segments in natural variations on the trend determination.

Furthermore, we have examined as thoroughly as possible the robustness of this identified speeding up in our replies to the reviewer's comments. This has been done by examining the uncertainty associated to the estimated trends calculated over different lengths of time (accounting for the autocorrelation in the noise residuals), by evaluating the need of additional measurements for confirming the trend and also by testing another approach (suggested by referee 2) that considers successive time segments of same length (over 6-year, 7-year and 8-years segments). These additional analyses have comforted our findings and this has been acknowledged by the referees. Note that we have specifically mentioned, at the end of Section 4.4, the need of additional measurements for verifying the trend derived over 2015-2017.

Please note also that in order to not oversell the manuscript or mislead the reader, we have decided to change the title of the manuscript to a question instead of a strong statement. Overall we believe that with these changes, which were suggested by the referees, we end with a careful and more balanced manuscript. Unless you object, we therefore would like to keep the paper as it is (the two referees also recommend to publish the paper "as is").

**Minor comments:**

Line 1-2: please indicate the time period in the title, "the first IASI decade (2008-2018) record"
Done as suggested: "Is the recovery of stratospheric O₃ speeding up in the Southern Hemisphere? An evaluation from the first IASI decadal record (2008-2017)"

Line 32: What is meant by "decline is not categorical in to total o3"?

By "not categorical", we mean "not unequivocal", according to the formalism of Tiao et al. (1990) and Weatherhead et al. (1998), as discussed in Section 4.

Line 32: "freezing the regression coefficient". What is this freezing?
We meant here that the regression coefficient were kept fixed, as explained in Section 4.4.

Line 36: "trends over the year"? Usually it's over a time period. Not for a year!
The term "trends over the year" means "annual trends", not "trends over one year".

Line 37: over the last years? How many years?
Over 2013-2017 as shown in Fig. 12 in Section 4.4. We believe there is no need to mention that in the abstract.

Line 39-40: it sounds that only IASI measurements are required to confirm the o3 change.
Our study presents the first detection of a speeding up of $O_3$ recovery process in the stratosphere over the whole year and estimates the number of additional measurements to assess a specified trend, based on the IASI dataset only.

Line 83: statistical approaches also use measurements or model results
The purpose of this sentence is to indicate that, similarly to our study, other studies already reported $O_3$ recovery from observational dataset using regression approach.

Line 97-101: The sentence is too hard to understand. Please split and rephrase it.
The sentence has been rephrased.

Line 103: what is high density here? Also join this small paragraph with the previous one.
This term is typically used for describing dataset with high spatio-temporal coverage and without major gap in the time series (e.g. as measured by IASI).
The paragraph has been joined to the previous one.

Line 110: Usually we use the acronym LS for the lower stratosphere, not LSt
We used that acronym to stay consistent with the previous IASI studies.

Line 119: "two years" is too short to estimate trends.
There is no two-year analysis in this paper (see the major comment). That sentence refers to the minimum numbers of years of IASI measurements that would be required to detect a specified trend.

Line 121: speeding up of o3 changes? Or speeding up of ozone recovery?
We mean the speeding up in the $O_3$ change rate. It might concern both the recovery and the decline.

Line 136: measurements of atmospheric composition? Or list some trace gases here.
"Measurements of atmospheric parameters" has been added.

Line 150: what is the vertical resolution of o3 profiles?

The resolution is variable along the vertical profile, but typically around ~6-8 km. As mentioned in Section 2.1, the vertical resolution allows to retrieve up to 4 independent levels of information on the vertical profile.

Line 158: Please state the accuracy of the retrievals
The accuracy is specifically discussed later in line 163-166.

Line 166: if 20% is the bias or accuracy, then how could that affect your statistical analyses presented, in terms of the ozone recovery estimation?
A so-called "bias" affects a dataset over the whole period of measurements, hence, it has no influence on the trend calculation. The distinction has to be made with the "drift" which describes a change in the bias and which has been specifically discussed in the manuscript (Section 2.1, L.168-188).

Line 177: "resulted from"
We believe "results from" is more appropriate here. The origin of the drift was not explained in the validation paper.

Line 181: full IASI period. Please state that particular period
The period is already mentioned in several places through the manuscript, e.g., title, abstract...

Line 186: is it the "trends" or any other "tendency"
By tendency, we referred to the trend. It has been changed to: "… trend observed in the IASI datasets".

Line 192: flags determined or selected
We choose to use the word "develop" because the IASI flags applied in this study have been specifically built to filter the IASI $O_3$ datasets (based on biased or sloped residuals, suspect averaging kernels, maximum number of iteration exceeded, fractional cloud cover …) as reported in previous IASI studies.

Line 303: it is a large range 25 to 95%. Could you please state where it reproduces the best and worst?
Done as suggested.

Line 322: delete "and discussed in this paper"
Done as suggested.

Line 368: Influence of QBO30 is greater, due to its altitude dependence?
The influence of the QBO is indeed altitude dependent. The lower influence of QBO10 vs QBO30 in the LSt would be explained by changes in the phase of the QBO10 response with destructive interference in the mid-low stratosphere as reported by Chipperfield et al. (1994) and Brunner et al. (2006) and mentioned in L.372 of the manuscript. Other couples of orthogonal functions could be used to account for both the strength and the phase of QBO (i.e. to adjust the time lag), but the QBO10 and 30 usually show the strongest anticorrelation.

Line 397: Yes, this has also been discussed in

*Roscoe, H. K. and Haigh, J. D.: Influences of ozone depletion, thesolar cycle and the QBO on the Southern Annular Mode, Q. J.Roy. Met. Soc., 133, 1855–1864, 2007.* and in Kuttippurath et al., 2013 (you have already cited this publication in this article).
Thank you for these additional references. They have been added.

Line 437-442: Please split this sentence, too long to comprehend
The sentence has been split.

Line 560: I do not understand this 80 DU here.
80 DU represents here the amplitude of the $O_3$ variability captured by IASI in the LSt during the austral spring in the highest southern latitudes, as illustrated in Fig.7a.

Line 620: 0.90; is it 90% CI level
"0.90" corresponds to the probability to detect a trend of a specified magnitude given at the 95% confidence level according to the trend detection formalism of Tiao et al. (1990) and Weatherhead et al. (1998). It does not refer to the confidence interval.
Note that the 95% confidence interval associated with the number of years required is also given in the manuscript (e.g. "detectable from ~2020-2022 ± 6-12 months"). This is now better explained in the manuscript.

Line 674: A correction: Kuttippurath and Nair (2017) discussed the vertical trends in ozone, not only total column ozone. In addition, they also discussed the ozone trends in summer months. A recent study from the same authors reported recovery signatures even at the saturation altitudes (doi:10.1038/s41612-018-0052-6).
The sentence now refers to the summer period as well and the reference Kuttippurath et al. (2018) has been added in the manuscript as suggested.

Line 805: You were discussing about 90% in previous pages. Now its 95%?
There is a difference between the 95% confidence level associated with the trend estimation and the probability of 90% to detect a specified trend (here given at that 95% confidence level).
Through the whole manuscript, as indicated in L216 in Section 2.2, the uncertainties associated to the trend values or to any other regression coefficients are given in the 95% confidence level, i.e. 2-sigma level.
Only the detection year for a specified trend is calculated within a probability of 90% according to the formalism of Tiao et al. (1990) and Weatherhead et al. (1998).

Line 808: Only 2- years are required? What is the basis for this statement?
It refers to the "additional years" of IASI measurements that are required to derive significant trends in the MUSt, as calculated from the Tiao et al. (1990) and Weatherhead et al. (1998) formalism in L. 621 (Section 4.1) of the revised manuscript.

We thank the Editor and the two referees for noting all the English style/technical/typos which have been corrected through the manuscript.